# Non-human primate model of long-COVID identifies immune associates of hyperglycemia

Clovis S. Palmer [1,2,9] ✉, Chrysostomos Perdios [1,2,9],
Mohamed Abdel-Mohsen [3], Joseph Mudd[1,2], Prasun K. Datta[1,2],
Nicholas J. Maness [1,2], Gabrielle Lehmicke[1], Nadia Golden [1], Linh Hellmers[1],
Carol Coyne[1], Kristyn Moore Green[1], Cecily Midkiff[1], Kelsey Williams [1],
Rafael Tiburcio[4], Marissa Fahlberg[1], Kyndal Boykin[1], Carys Kenway[1],
Kasi Russell-Lodrigue [1,5], Angela Birnbaum[1,8], Rudolf Bohm[6], Robert Blair [1,7],
Jason P. Dufour[1,5], Tracy Fischer [1,2], Ahmad A. Saied [1,7] & Jay Rappaport [1,2] ✉

Hyperglycemia, and exacerbation of pre-existing deficits in glucose metabolism, are manifestations of the post-acute sequelae of SARS-CoV-2. Our understanding of metabolic decline after acute COVID-19 remains unclear due to the lack of animal models. Here, we report a non-human primate model of metabolic post-acute sequelae of SARS-CoV-2 using SARS-CoV-2 infected African green monkeys. Using this model, we identify a dysregulated blood chemokine signature during acute COVID-19 that correlates with elevated and persistent hyperglycemia four months post-infection. Hyperglycemia also correlates with liver glycogen levels, but there is no evidence of substantial long-term SARS-CoV-2 replication in the liver and pancreas. Finally, we report a favorable glycemic effect of the SARS-CoV-2 mRNA vaccine, administered on day 4 post-infection. Together, these data suggest that the African green monkey model exhibits important similarities to humans and can be utilized to assess therapeutic candidates to combat COVID-related metabolic defects.

Between 10 and 30% of people infected with SARS-CoV-2 develop long-term health complication, named post-acute sequelae of SARS-CoV-2 (PASC) or Long-COVID[1–4]. Metabolic diseases, including type 2 diabetes (T2D)[5], as well as conditions with less obvious metabolic undertones such as myalgic encephalomyelitis/chronic fatigue syndrome (ME/CFS), breathlessness, thrombosis and neuropsychiatric sequelae (brain fog) embody the broad spectrum of PASC[1,6–11]. Compared to uninfected controls, individuals infected with SARS-CoV-2 have elevated risks and 12-month burdens of diabetes[12]. Several lines of evidence suggest a hyperinflammatory response against SARS-CoV-2 as being critical to the severity of acute COVID-19[13], as well as the development of metabolic PASC such as hyperglycemia[3,12], metabolic associated fatty liver disease (MAFLD)[14], and cardiovascular diseases[15,16].

An existing paradigm postulates that the balance between virus survival and effective host responses is based on the metabolic reprogramming of nutrients, primarily in immune cells[17]. However, the extent to which early disruptions in systemic immune and metabolic homeostasis contribute to the evolution and symptoms of metabolic

[1]Tulane National Primate Research Center, Covington, LA, USA. [2]Department of Microbiology and Immunology, Tulane University School of Medicine, New Orleans, LA, USA. [3]The Wistar Institute, Philadelphia, PA, USA. [4]Division of Experimental Medicine, Department of Medicine, University of San Francisco, CA, USA. [5]Department of Medicine, Tulane University School of Medicine, New Orleans, LA, USA. [6]Oregon National Primate Research Center, Oregon Health and Science University, Beaverton, OR, USA. [7]Department of Pathology and Laboratory Medicine, Tulane University School of Medicine, New Orleans, LA, USA. [8]Deceased: Angela Birnbaum. [9]These authors contributed equally: Clovis S. Palmer, Chrysostomos Perdios. ✉e-mail: cpalmer3@tulane.edu; jrappaport@tulane.edu

PASC remains unclear. While pre-existing diabetes has been linked to more severe COVID-19 outcomes and higher mortality[18,19], new-onset hyperglycemia and diabetic ketoacidosis are also observed in SARS-CoV-2-infected individuals with no prior evidence of diabetes and have been associated with poor COVID-19 outcomes[20,21].

Glucose homeostasis is maintained largely by hormonally-regulated glucose uptake by tissues, such as the liver[22,23], as well as the gut microbiota[24]. However, hepatic glucose production via gluco-neogenesis and glycogenolysis represents additional glucometabolic checkpoints[25]. In addition, β-cell dysfunctions, potentially due to early pancreatic infection by SARS-CoV-2, may partially contribute to altered glucose homeostasis[26–28]. Increased levels of circulating inflammatory molecules, including chemokines, have been shown to directly associate with impaired glucose homeostasis in non-infectious diseases[29]. This includes CCL25, a cytokine known to impair pancreatic β-cell insulin secretion and can induce proinflammatory cytokine responses[30]. Thus, a preponderance of evidence supports key role for inflammation in the pathogenesis of hyperglycemia and T2D[31–33]. However, the mechanisms by which SARS-CoV-2 infection promotes prolonged hyperglycemia are poorly understood due to the lack of appropriate animal models for metabolic PASC. Here, we developed such a model, by utilizing SARS-CoV-2-infected non-human primates (NHPs). We analyzed blood biochemistry, virologic, and immunologic parameters longitudinally over an 18-week study period. In addition, metabolic, virologic, and clinical analyses on selected tissues at necropsy were used to interrogate potential mechanisms that underlie the development of metabolic PASC.

Intrahepatic inflammation caused by viruses (e.g., Hepatitis C Virus)[34], as well as SARS-CoV-2 infection of the pancreas[35] has been linked to diabetes onset. We reasoned that vaccination against SARS-CoV-2 during acute infection prior to multiorgan distribution from the lungs, may elicit a more favorable tissue immune microenvironment that reduces the tissue burden of replication competent SARS-CoV-2, and or viral fragments that may aggravate tissue inflammatory and metabolic dysfunction. We therefore investigated whether administration of the BNT162b2 (Pfizer/BioNTech) vaccine four days post SARS-CoV-2 infection could ameliorate immunometabolic dysregulation.

In this work, we report SARS-CoV-2-infected African green monkeys (AGMs) exhibit persistent hyperglycemia up to four months post infection. We identify a plasma chemokine signature during acute COVID-19 that correlates with this metabolic defect. Hyperglycemia also correlates with inflammatory T-cell populations and hepatic glycogen levels. Finally, we report a favorable metabolic effect of the SARS-CoV-2 mRNA vaccine, administered on day 4 post infection. These data suggest SARS-CoV-2-infected AGMs model exhibits important features with human metabolic PASC and can be explored to assess therapeutic candidates for PASC.

## Results
### Characteristics of study groups
African green monkeys (AGMs; *Chlorocebus aethiops sabaeus*) were infected with SARS-CoV-2 (strain 2019-nCoV/USA-WA1/2020) intranasally at ~1e6 TCID50 (0.5 mL/nares), and intratracheally (1 ml). The cohort was followed up weekly for 18 weeks with complete virologic, physical, clinical assessments, blood chemistry, and immunometabolic profiling. The cohort was split into two main groups. The unvaccinated group (*n* = 10) got no vaccination against SARS-CoV-2, and the vaccinated group (*n* = 5) was vaccinated 4 days post infection (dpi) with the mRNA vaccine BNT162b2 (Pfizer/BioNTech). Some assessments were conducted biweekly after 4 weeks (Fig. 1a). One female (PB24; vaccinated), age 19.32 years (6.40 kg) was sent for necropsy at week 8 due to anorexia. There were no significant differences in the median age and weight between the vaccinated and unvaccinated group (see Table 1 for detailed group demographics). Blood/plasmas were collected in the mornings from all 15 animals (fasted overnight) at baseline 11 days

pre-infection unless otherwise indicated, and longitudinally at day 3, and weeks 1, 4, 8, 12, 16, and 18 post-infection (p.i.), unless otherwise indicated, for virologic, blood chemistry, cytokine/chemokine, and immune/antibody response analyses. Pre-baseline levels of blood glucose were measured every few months over a three-year period. Liver, pancreas, duodenum, and lung samples collected at necropsy (18 weeks p.i.) were used for virologic, metabolic, and clinical evaluations.

### Dynamics of SARS-CoV-2 over time
To confirm infection and study viral dynamics we assessed sub-genomic viral RNA (sgRNA), a correlate of actively replicating virus, and genomic RNA (gRNA) levels by quantitative real-time PCR (viral copies (VC)/ml) in nasal and pharyngeal swabs. All animals had detectable sgRNA ($>3.64 \times 10^6$) at 3 dpi in both nasal and pharyngeal swabs. At week 5, only 2 animals had detectable sgRNA ($>1.15 \times 10^5$) in nasal swabs, and no detection in pharyngeal swabs. At day 3, all animals had detectable gRNA ($>2.05 \times 10^7$) in nasal swabs, while 14 had detectable gRNA ($>1.71 \times 10^6$) in pharyngeal swabs. At week 5, seven animals had detectable gRNA ($1.47 \times 10^5$–$6.53 \times 10^6$) in nasal swabs, and seven detectable ($4.90 \times 10^4$–$1.51 \times 10^5$) in pharyngeal swabs. The kinetics of gRNA were similar in vaccinated and unvaccinated, with virus peaking at day 3 and substantially declining by week 5 (Fig. 1b). At week 5, 20% and 60% of the vaccinated and unvaccinated animals, respectively, had detectable gRNA in pharyngeal swabs (Fig. 1b, pie charts).

### Changes in immune compartments, antibodies, and ex-vivo polyclonal activation profiles during long-term follow-up
To evaluate changes in systemic immune cell compartments over time, we assessed major immune subsets that have previously been implicated in the severity of acute COVID-19 disease. On day 3 p.i., we observed a significant increase in monocyte percentage ($p = 0.0004$) and absolute number ($p = 0.002$) in infected animals, returning to baseline levels by week two, followed by a modest decline up to about week 7 (Fig. S1a, b). The percentage and absolute counts of lympho-cytes and neutrophils remained relatively constant (Fig. S1c–f) in both the vaccinated and unvaccinated groups. The declining levels of cir-culating monocytes post-acute infection may signify potential infil-tration of these cells into tissues, consistent with elevated tissue inflammation during acute SARS-CoV-2 infection.

We observed a steep induction of IgG and IgA responses against SARS-CoV-2 Spike and S1RBD proteins, peaking between 3–6 weeks (Fig. 1c, d), and maintained elevated up to 18 weeks p.i. in both the vaccinated and unvaccinated groups. There was also an appreciable IgA and IgG response against SARS-CoV-2 nucleocapsid, with the IgG levels remaining elevated up to 15 weeks (Fig. 1e). The early IgM response against SARS-CoV-2 was noticeable but of a lesser magnitude than the IgA and IgG responses, and sharply declined towards baseline after 5-week p.i. (Fig. 1c–e). Although not reaching statistical sig-nificance, cumulatively, there were substantially elevated IgG and IgA antibody responses against the Spike (Fig. 1f, g) and S1RBD (Fig. 1h, i) proteins at week 3 p.i. relative to baseline (1.5 weeks pre-infection). These results closely reflect the kinetics and preferential induction of anti-SARS-CoV-2 IgA, IgG, and IgM responses in infected humans[36].

There were no significant differences in the magnitude of antibody responses between the vaccinated and unvaccinated groups over time (PERMANOVA; $p > 0.05$, Fig. 1c–e). However, there was a slight trend towards a higher induction of IgA response towards the spike protein, as well as IgG response towards nucleocapsid protein (Fig. 1c, e).

The percentages of IFNγ-producing CD4+ and CD8+ T cells in response to in vitro exposure to polyclonal stimulants phorbol 12-myristate 13-acetate (PMA), and the calcium ionophore ionomycin (I) is increased in severe and extreme COVID-19 disease compared to mild[13]. To assess this response, we exposed longitudinally collected PBMCs

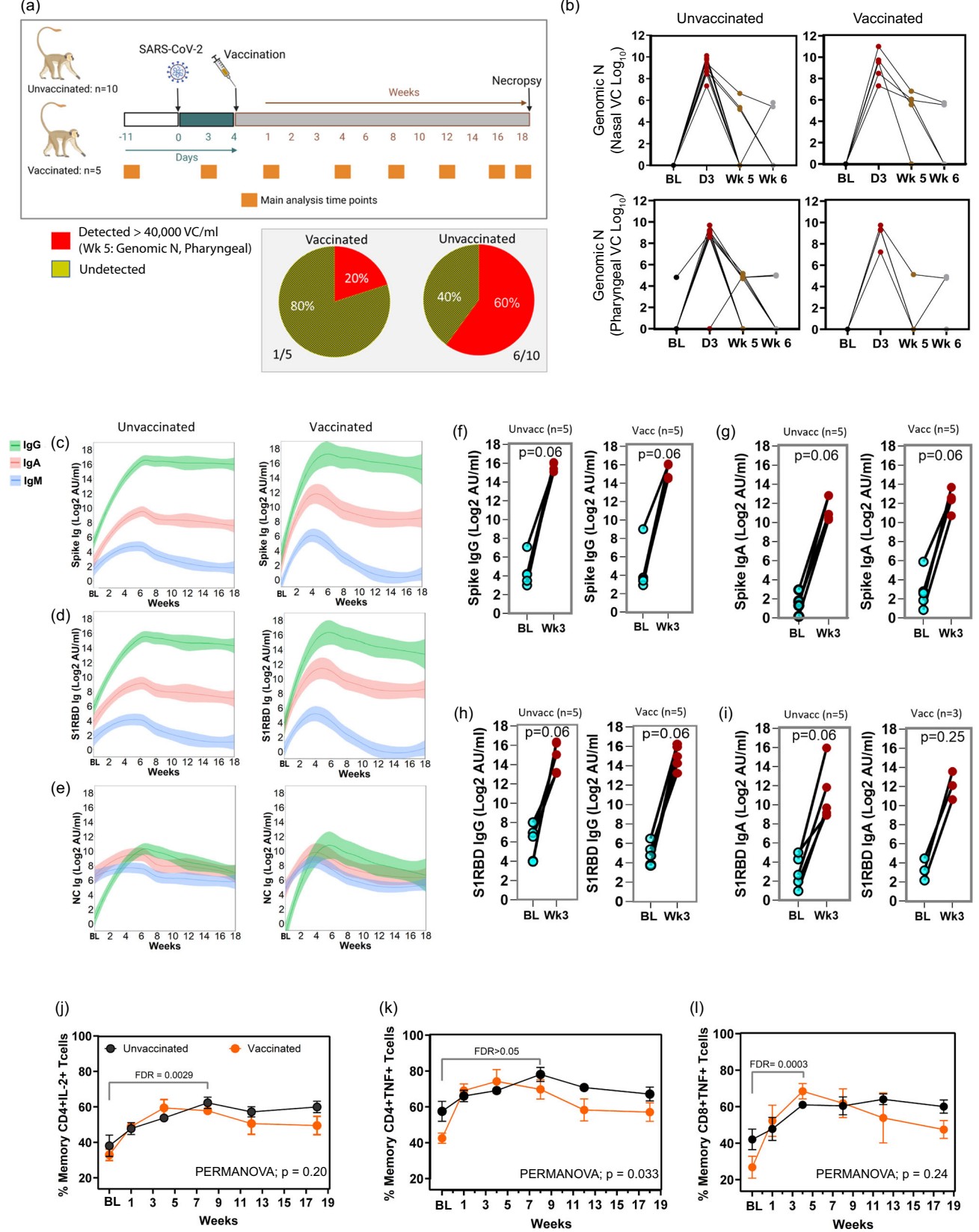

from SARS-CoV-2-infected AGMs to PMA/I for 6 h, examined activation markers and intracellular levels of cytokines in T cell subsets, and used gating in FlowJo to identify targeted T cell populations. Naïve and memory populations were defined using CD95 and CD28 markers according to established gating strategies (Fig. S2). Cumulatively, across groups, there was a significantly increased percentage of memory CD4+ T cells expressing IL-2 (Fig. 1j; FDR < 0.01) between baseline and all following weeks p.i. Cumulatively, across timepoints, there was a significantly lower percentage of memory CD4+ T cells expressing TNF in the vaccinated group versus the unvaccinated one

**Fig. 1 | SARS-CoV-2 infection of AGM reflects virologic, and immune response reported in humans. a** An overview of the study design and major analysis timepoints. Baseline (BL) samples are taken 11 days pre-infection except when otherwise indicated. Fifteen animals were infected with SARS-CoV-2 (strain 2019-nCoV/USA-WA1/2020) intranasally, and intratracheally. Five received the BNT162b2 (Pfizer/BioNTech) mRNA vaccine 4 days post-infection (dpi), while 10 remained unvaccinated against SARS-CoV-2. Main analysis timepoints (BL, day 3, and weeks 1, 4, 8, 12, 16, and 18 p.i.) for mainly blood chemistry, chemokines/cytokines, and immune responses are indicated. Tissues (liver, pancreas, duodenum, lung) were collected at necropsy (18 weeks p.i.) for virologic analysis, metabolic and or clinical evaluation. Figure 1a created with BioRender.com released under a Creative Commons Attribution-NonCommercial-NoDerivs 4.0 International license. **b** Nasal and pharyngeal SARS-CoV-2 genomic nucleocapsid (N) RNA in unvaccinated and vaccinated animals (right), and pie charts showing the percentage of animals with detectable pharyngeal gRNA in animals at week 5 (bottom left). $n = 10$ and 5 unvaccinated and vaccinated animals, respectively. **c** Plasma antibody responses against spike, **d** S1RBD, and **e** nucleocapsid (NC) proteins. **f** Timepoint analysis of IgG, and **g** IgA responses against spike protein. **h** Timepoint analysis of IgG, and **i** IgA responses against S1RBD protein. **j** Percentage of memory CD4+ T cells expressing IL-2. **k** The percentage of memory CD4+ T cells expressing TNF in response to PMA/ionomycin (P/I) stimulation. **l** The percentage of memory CD8+ T cells expressing TNF in response to P/I stimulation. For **j–l** unvaccinated: BL, $n = 8$; Wk 1–18, $n = 10$; Vaccinated: BL-Wk 8 and Wk 18, $n = 5$; Wk 12, $n = 4$. Two-sided Wilcoxon matched-pairs signed-rank test with Benjamini–Hochberg correction (FDR) was used to compare timepoints. PERMANOVA was used for temporal comparisons between groups. BL: 1.5 weeks pre-infection. **c–e** shade represents 95% confidence interval. **j–l** error bars represent SEM. $n$ denotes number of independent samples/animals at each timepoint. Source data are provided as a Source Data file.

(Fig. 1k; PERMANOVA; $p = 0.033$). Except for week 1 (FDR > 0.05), overall, the percentage of memory CD8+ T cells expressing TNF was significantly increased relative to baseline (Fig. 1l; FDR < 0.01). Taken together, these data indicate that the AGM SARS-CoV-2 infection model mirrors several immunological similarities to those reported in SARS-CoV-2-infected humans.

## Vaccination post SARS-CoV-2 infection improves long-term glycemic control

Early metabolic changes during SARS-CoV-2 infection are likely to influence long-term manifestations of COVID-19. We have previously shown in AGMs that SARS-CoV-2 infects pancreatic ductal, and endothelial cells and associates with new-onset hyperglycemia[35], an observation that recapitulates findings in humans, supporting pancreatic tropism of SARS-CoV-2[27,28]. SARS-CoV-2 infection may also promote hyperglycemia by inducing excess hepatic glucose production through gluconeogenesis or glycogenolysis independent of pancreatic function or insulin effects[37]. We therefore analyzed serum glucose levels over time and found it to be significantly elevated ($n = 15$, mean: 102.2 mg/dL, range: 76–154, $p < 0.0001$) as early as three days post infection relative to the latest baseline values ($n = 15$, mean: 71.1 mg/dL, range: 51-86; Fig. 2a). This elevated blood glucose is well above the normal range independently reported for male (mean: $81.7 \pm 18.0$ mg/dL) and female (mean: $80.3 \pm 18.7$ mg/dL) AGMs of Caribbean origin[38]. Composite longitudinal analysis showed significantly higher blood glucose in the unvaccinated group (PERMANOVA; $p = 0.001$) than the vaccinated group over time. This is supported by a higher percentage of animals with glucose levels above 100 mg/dL in the unvaccinated group (Fig. 2b). Moreover, a statistically significant persistence of hyperglycemia was maintained up to at least 12 weeks p.i. in the unvaccinated group (Fig. 2c). We excluded glucose reading for weeks 10 and 18 due to in-house clinical procedures likely to impact transient blood levels, namely the movement of animals from BSL3 to BSL2 (week 10) and return to BSL3 at week 18. There was a modest positive but non-statistically significant relationship between day 3 nasal viral sgRNA and gRNA, and blood glucose levels at week 17

across groups (Fig. 2d). Peak IgA response (week 3) against nucleocapsid and S1RBD was lower in animals with higher preceding blood glucose (week 2; Fig. 2e).

We retrospectively examined the pre-baseline glucose levels in blood collected every few months in a three-year period (December 2018–November 2021) from animals housed in the colony, that were then transferred to BSL3/2 during the study period. The glucose levels over this period were relatively consistent, with no significant differences in the cumulative values between animals that were marked for the vaccinated or unvaccinated group (Fig. S3a–c). Of the 13 timepoints, from week 4 to week 17 post SARS-CoV-2 infection, blood glucose levels were significantly different from cumulative pre-baseline levels 9 time in the unvaccinated group (9/13), versus only two time in the vaccinated group (2/13; Fig. S3d, e). Moreover, the cumulative pre-baseline glucose level was significantly lower than the composite blood glucose readings from week 4 to week 17 p.i. in the unvaccinated group (Fig. S3f). Although serum triglyceride levels peaked around week 2, the levels were already significantly elevated by week 1 in the unvaccinated group (Fig. 2f, g). Cumulatively, triglyceride levels were significantly higher in the unvaccinated group over time (PERMANOVA; $p = 0.001$). However, this elevation was not maintained consistently beyond baseline over the study period (Fig. 2f, g). Except between baseline and week 1, we recorded no significant composite difference in cholesterol levels between the two groups over time (Fig. 2h, i). Taken together, we reveal a significant induction and persistent hyperglycemia in SARS-CoV-2-infected AGMs, suggesting this experimental design as a potential model of metabolic PASC. Furthermore, we found that vaccination during acute infection could have a positive effect on glycemic control and may be explored to reduce the long-term burden of new-onset COVID-19-related diabetes.

## Induction of early inflammatory disturbances in AGM long-COVID model

We next undertook an unbiased approach to identify systemic cellular processes associated with SARS-CoV-2 infection in AGMs. We utilized the OLINK Proximity Extension Assay (PEA) technology, which has exceptional specificity requiring the binding of two matched-paired antibodies tagged with a unique DNA sequence followed by PCR amplification and signal generation. We used a Target 96 Inflammation Panel assessing proteins associated predominantly with apoptosis, immune activation and inflammatory responses, MAPK cascade, chemotaxis, and chemokine secretion. Sixty-five analytes were analyzed following stringent data QC of which 14 were significantly elevated, and one (IL-8) significantly reduced at week 1 p.i. (Fig. 3a, Table S1). Principal component analysis of these differentially regulated analytes expectedly showed a marked separation between baseline and week 1, corroborating the heatmap analysis (Fig. 3a, Table S1). The 14 elevated analytes include some with known chemotaxis and other inflammatory processes. Five analytes, C-C Motif Chemokine Ligand 25 (CCL25), CUB

**Table 1 | Characteristics of the study population, and the vaccinated and unvaccinated subpopulations**

| Characteristics | Study population | Unvaccinated | Vaccinated (Pfizer) | p |
|---|---|---|---|---|
| N | 15 | 10 | 5 | |
| Sex (F/M) | 13/2 | 9/1 | 4/1 | |
| Age (years), median (min-max) | 8.97 (7.92–19.32) | 8.97 (7.92–17.92) | 8.97 (7.92–19.32) | 0.56 |
| Weight (kg) (min–max) | 4.35 (3.75–6.55) | 4.35 (3.75–5.45) | 4.60 (4.15–6.55) | 0.29 |

Statistical comparison between groups was done using the two-sided Mann–Whitney U test.

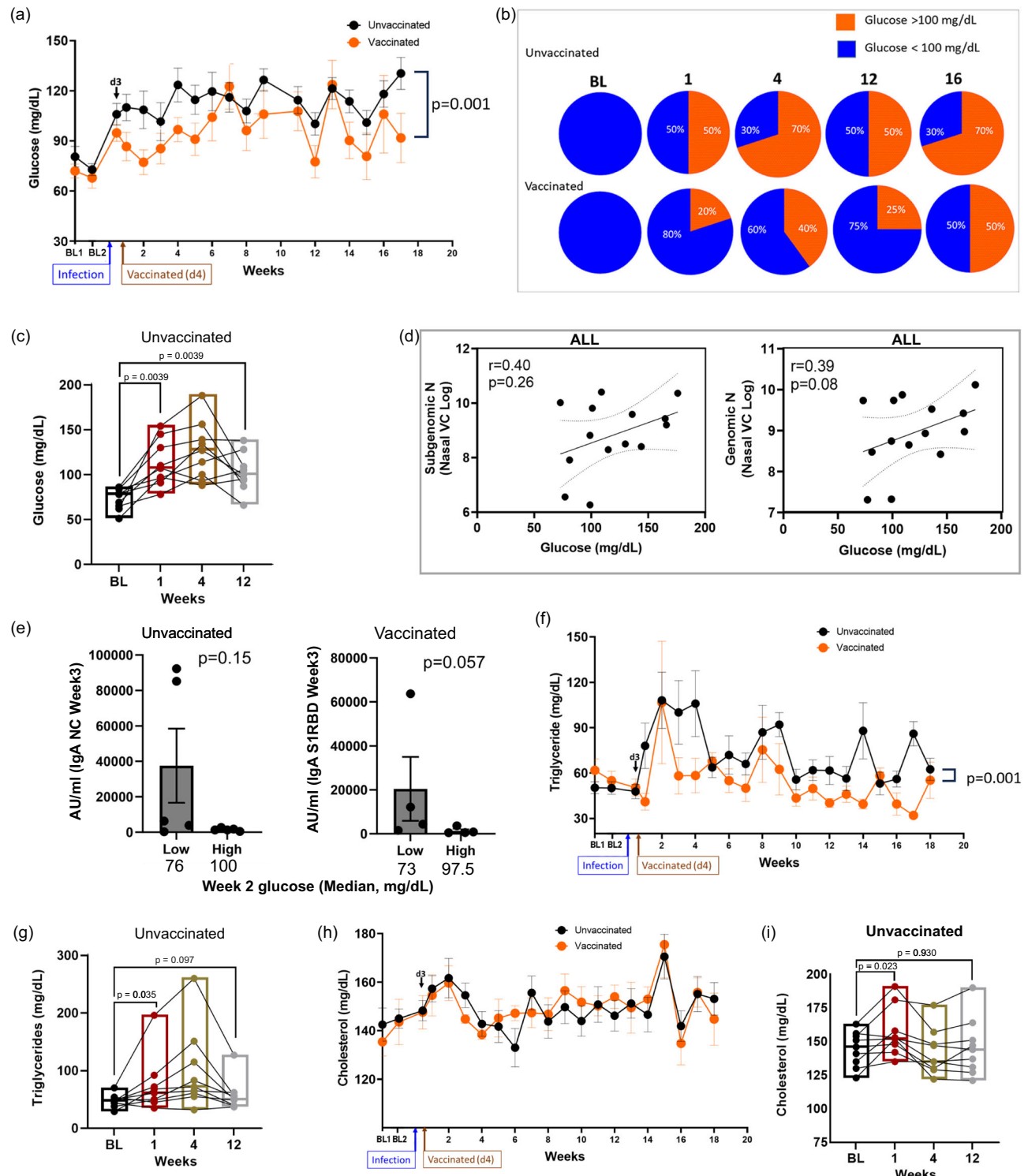

**Fig. 2 | SARS-CoV-2 infection of AGM is associated with elevation and persistence of blood glucose concentration even after undetectable virus. a** Levels of plasma glucose following infection over time. PERMANOVA analysis was used for statistical analysis to examine the cumulative difference between glucose levels in the two groups over time. Unvaccinated: $n = 10$; Vaccinated: BL-Wk 8, $n = 5$; Wk 9-Wk 17, n-4. **b** Pie charts showing the proportion of animals with glucose >100 mg/dL at baseline (BL) and week 1, 4, 12 and 16. Unvaccinated: $n = 10$; Vaccinated: BL-Wk 4, $n = 5$; Wk 9-Wk 16, $n = 4$. **c** Glucose levels at BL and week 1, 4, 12 in unvaccinated animals ($n = 9$) post infection (p.i.), analyzed using Two-tailed Wilcoxon matched pairs signed-rank test. **d** Two-sided Spearman correlation between glucose levels and subgenomic nucleocapsid (N) and genomic N at day 3 post infection in all animals. **e** IgA response against NC ($n = 5$) and S1RBD ($n = 4$) at Wk 3 against unvaccinated animals with low and high glucose in the preceding week

(week 2), analyzed using Two-tailed Man Whitney $T$ test. **f** Triglyceride levels over time between to the vaccinated and unvaccinated group. Unvaccinated: $n = 10$. Vaccinated: BL-Wk 8, $n = 5$; Wk 9-18, $n = 4$. PERMANOVA was used for statistical analysis. **g** Triglyceride levels at BL and week 1, 4, 12 in unvaccinated animals ($n = 9$) p.i., analyzed using Two-tailed Wilcoxon matched pairs signed-rank test. **h** Total cholesterol levels between the unvaccinated and vaccinated group. Unvaccinated: $n = 10$. Vaccinated: BL-Wk 8, $n = 5$; Wk 9-18, $n = 4$. **i** Cholesterol levels at BL and week 1, 4, 12 in unvaccinated animals ($n = 9$) p.i., analyzed using Two-tailed Wilcoxon matched pairs signed-rank test. Key: BL1 = baseline 1 (6.5 weeks pre-infection); BL2 = baseline 2 (11 days pre-infection); d3 = day 3. PERMANOVA was used to compare groups in **a**, **f**. Error bars represent SEM. $n$ denotes number of independent samples/animals at each timepoint. Source data are provided as a Source Data file.

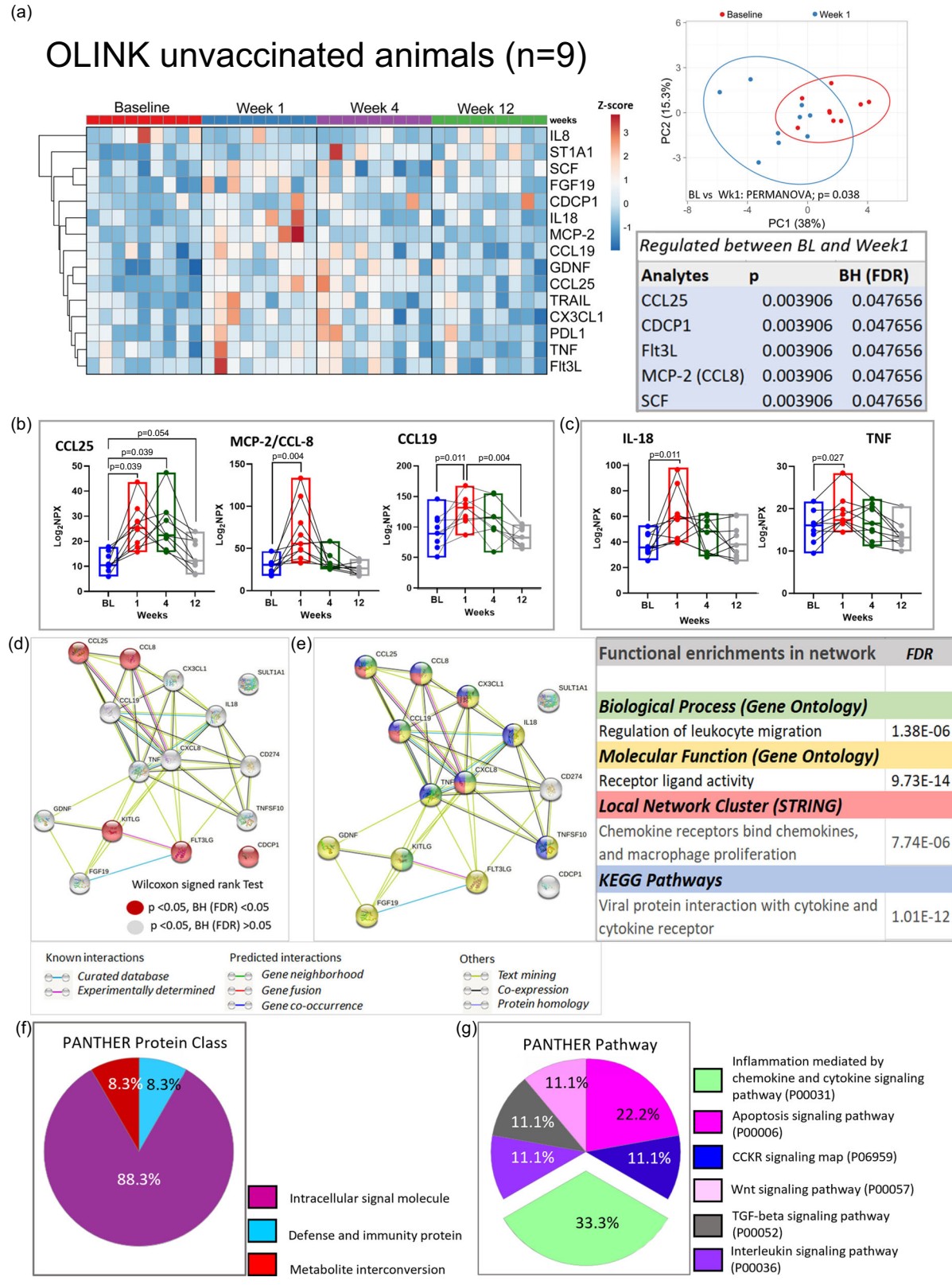

domain-containing protein 1 (CDCP1), FMS-like tyrosine kinase 3 ligand (Flt3L), C-C Motif Chemokine Ligand 8 (CCL8) and Stem Cell Factor (SCF) maintained significance (FDR < 0.05) following Benjamini–Hochberg (BH) correction (Table insert, Fig. 3a). We highlight representative chemokines (CCL25, CCL8, CCL19) and inflammatory molecules (IL-18, TNF) showing significant elevation at week 1, normalizing by week 12 (Fig. 3b, c).

We next examined whether these 15 differentially regulated analytes were interrelated or shared common pathways. We employed the STRING protein-protein analysis web-based tool using the default settings. Except for CDCP1 and SULT1A1 (ST1A1) there were high confidence interconnections between the regulated analytes, notably the chemokines CCL8, CCL19, CCL25, CXCL8, CX3CL1, and TNF and IL-18 (Fig. 3d). The top functional enriched networks revealed biological

**Fig. 3 | Elevated inflammatory mediators during acute SARS-CoV-2 infection of unvaccinated AGMs. a** Heatmap depicts 15 plasma analytes dysregulated between baseline and week 1 during SARS-CoV-2 infection of 9 AGMs ($p < 0.05$). Heat colors show standardized Z scores; red indicates the highest levels of analytes, and blue indicates the lowest levels. PCA plot confirms significant differences in analyte expression at baseline (red) and week 1 (blue). Box insert shows 5 dysregulated analytes remaining significantly regulated after Benjamini–Hochberg (BH) correction (FDR < 0.05). **b** Selected chemokines and **c** inflammatory markers upregulated in plasma of unvaccinated AGMs ($n = 9$) following SARS-CoV-2 infection. Statistical comparison between time points was performed using a two-tailed Wilcoxon matched-pairs signed-rank test. **d** STRING protein-protein interaction network based on the 15 significantly differentially regulated plasma proteins at week 1.

Proteins significantly regulated at $p < 0.05$ with FDR > 0.05 are highlighted in gray. Those with FDR < 0.05 are highlighted in red. **e** Top functionally enriched networks associated with the 15 regulated plasma proteins in STRING. Proteins in biological processes associated with the regulation of leukocyte migration are highlighted in green. Proteins with molecular functions associated with receptor-ligand activities are highlighted in yellow. Proteins associated with chemokine receptor binding, and macrophage proliferation are highlighted in red. Proteins in KEGG pathways associated with the regulation of leukocyte migration are highlighted in blue. Proteins that entered the network based on close associations are denoted by gray. **f** PANTHER 17.0 classification of differentially regulated proteins based on protein class, and **g** pathway. $n$ denotes the number of independent samples/animals at each time point. Source data are provided as a Source Data file.

processes and molecular functions associated with regulation of leukocyte migration and receptor ligand activity. The top-ranked local network clusters, and KEGG pathways were related to chemokine receptors, macrophage proliferation, and viral protein interaction with cytokines and cytokine receptors (Fig. 3e, Table S2). We validated these networks using PANTHER v 17.0 revealing the protein class was dominated by intracellular signal molecules (Fig. 3f), with top pathways associated with inflammation mediated by chemokine and cytokine signaling, and apoptosis (Fig. 3g), supporting the STRING analysis. These data identify a specific dysregulation in inflammatory markers in the AGM metabolic PASC model.

### Glial cell line-derived neurotrophic factor (GDNF) remains persistently elevated following SARS-CoV-2 infection of AGM and is associated with hyperglycemia

Increased serum concentrations of inflammatory cytokines correlate positively with fasting glucose levels in individuals with features of the metabolic syndrome[39]. We questioned whether early pathological inflammatory processes may be associated with early-onset, or persistence of hyperglycemia in our SARS-CoV-2-infected AGM long-COVID model. We conducted correlation analysis of blood glucose levels at week 1, 4, 8, and 16 against all analytes that were significantly (raw $p$ value < 0.05) differentially regulated in the 10 unvaccinated animals at week 1 (Fig. 4a). We observed a modest but statistically significant correlation ($p < 0.05$) between week 1 plasma CDCP1 levels and week 4 and 16 glucose levels, and a significant and strong positive correlation between week 1 plasma GDNF and week 4 glucose levels (Fig. 4a, Fig. S4). We explored the overall relationship between total plasma analytes and changes in serum glucose concentrations over time in the unvaccinated group. We identified 8 analytes CCL25, GDNF, ADA, ST1A1, CXCL9, IL-10RB, FGF-19 and CDCP1 that positively and significantly correlated with plasma glucose across baseline, week 1, 4, and 12, and one analyte, IL-8, which had a significant negative association with glucose. CCL25 ($r = 0.57$; $p = 0.0003$; FDR = 0.004), GDNF ($r = 0.55$; $p = 0.0004$; FDR = 0.006) and ADA ($r = 0.44$; $p = 0.007$; FDR = 0.049) maintained significance following BH correction (Table S3). The CCL25 and GDNF data are graphically represented (Fig. 4b). STRING functional protein interaction analysis of these 9 analytes identified CXCR chemokine receptor binding (FDR = 0.019), chemokine receptor chemokines (FDR = 0.009), and viral protein interaction with cytokines and cytokine receptors (FDR < 0.0001) as top functional enrichments networks.

Of these analytes GDNF levels remained persistently and significantly higher above baseline up to week 12 p.i. (Fig. 4c), and correlated positively and significantly with plasma CCL25 ($r = 0.64$, $p < 0.0001$), a chemokine linked to T2D[30] (Fig. 4d). We employed a confirmatory dataset comprising of plasma from both unvaccinated and vaccinated animals at baseline and 1–3 weeks and evaluated GDNF levels using the OLINK panel indicated above. We confirmed significantly increased plasma GDNF following SARS-CoV-2 infection of AGMs. At the same timepoints, there was no significant elevation in plasma GDNF in

the vaccinated group (Fig. 4e). Interestingly, there was no significant correlation between blood insulin levels and glucose (Fig. 4f).

To gain more insights into the functionality of GDNF, we conducted unbiased analysis by using GDNF as the only search variable in STRING and set the interaction stringency to highest allowable confidence (0.90). Based on experimental evidence, databases, and text mining the analysis map revealed a high confidence interaction (0.999) between GDNF and its receptors GFRA1, and GFRA2, and the putative receptor RET receptor tyrosine kinase, which is involved in neuronal navigation and cell migration. There were also high confidence interactions between GDNF and neural cell adhesion molecule 1 (NCAM1; 0.936), a cell adhesion protein involved in neuron-neuron adhesion, and between GDNF and cyclic AMP-responsive element-binding protein 1 (CREB1), a transcription factor activated upon binding to the DNA cAMP response element (CRE) found in many viral promoters (Fig. 4g).

Finally, we used BioGRID, another protein-protein interaction database, to validate STRING's GDNF interactions. The results confirmed our STRING analysis, showing high-confidence interactions between GDNF and its receptors (red arrows), as well as with WDR26 (evidence: affinity capture-MS; green arrow; Fig. 4h), which serves as a scaffold to coordinate PI3K/Akt activation[40], as well as regulating leukocyte migration[41]. GDNF also interacts strongly with epidermal growth factor receptor (evidence: affinity capture-MS; green arrow) reported to regulate the severity of COVID-19 in patients[42]. Together, these results identify a potentially early and minimal host inflammatory signature dominated by elevation of plasma chemokines and the neurotropic factor GDNF, associated with impaired glucose homeostasis, in SARS-CoV-2-infected AGM.

### Increased percentage of inflammatory cytokine-producing CD8+ T cells in response to ex-vivo polyclonal stimulation positively correlates with blood glucose levels in SARS-CoV-2-infected AGMs

As shown above from our targeted gating, increased T cell activation and cytokine production in response to polyclonal stimulation is elevated following SARS-CoV-2 infection in AGMs. We then set out to explore unique populations of PMA/I responding T cells to determine whether the level of responsiveness to polyclonal stimulation correlates with blood glucose levels in infected AGMs. PBMCs were exposed to PMA/I for 6 h and the levels of T cell activation (CD69 marker) and cytokine production were monitored by flow cytometry. We imported data in FlowJo, gated on singlets, live cells and CD45+ CD3+ T cells (Fig. S2) and exported CSV–Scale values. We used the R package Spectre to identify 18 CD3+ populations of interest (Fig. S5a) at baseline, week 1, 4, 8, and 12 in PBMC samples. A composite UMAP derived from unstimulated (Fig. 5a), and PMA/I stimulated (Fig. 5h) samples is shown, with additional UMAPS displaying the normalized expression patterns of selected cell surface (CD8, Fig. 5b), activation (CD69, Fig. 5c), and inflammatory markers (TNF, IFNy, IL-2) in untreated (Fig. 5d–f) as well as stimulated cells (Fig. 5j–m). Expression patterns of

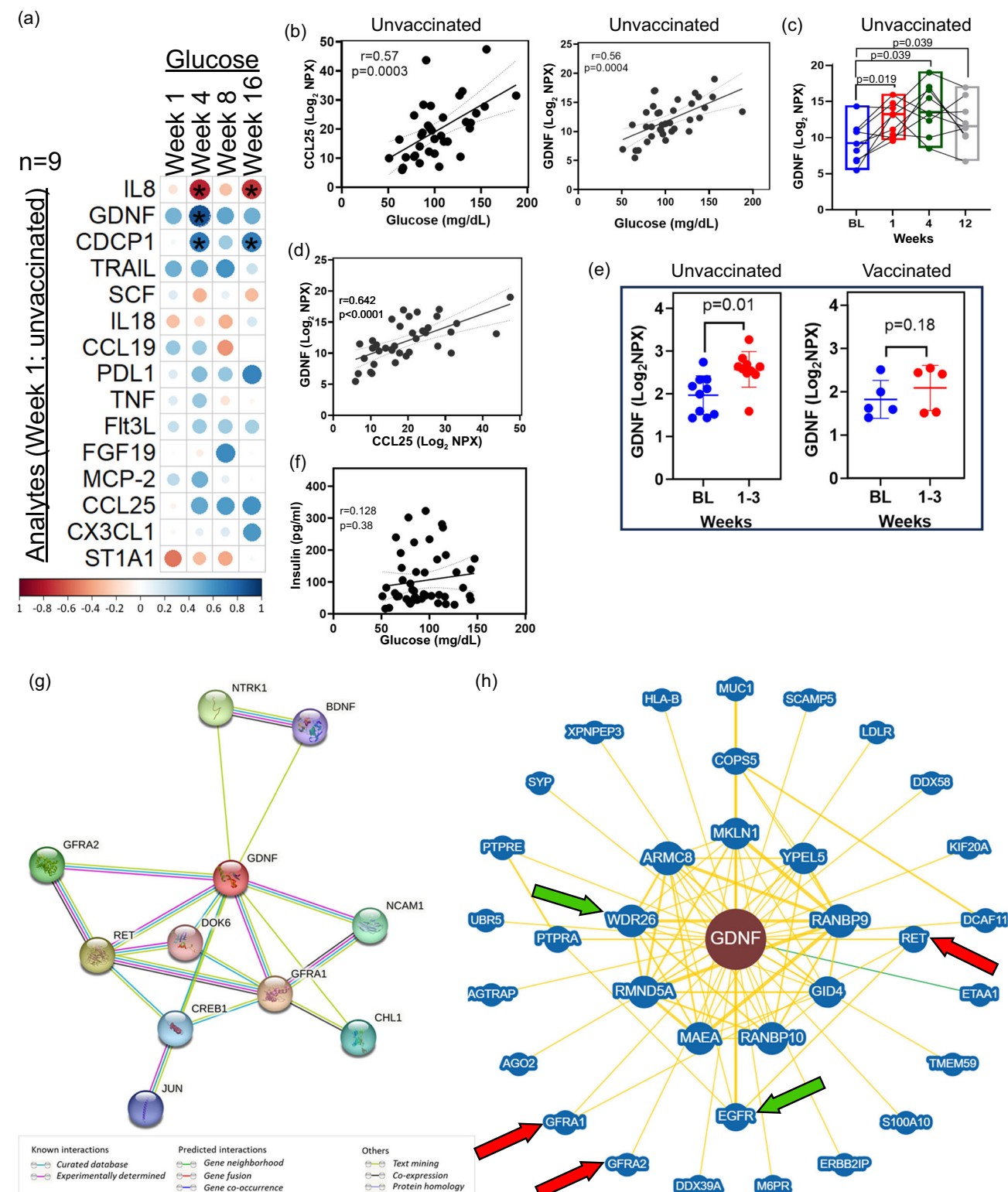

maturation markers CD28 and CD95 are shown (Fig. S5b). A density UMAP map plot (Fig. 5g, n) highlights cell density among the different clusters. The high expression (deep red) of activation and inflammatory markers confirmed T cell activation by PMA/I (Fig. 5j–m). Across all animals over the 5 timepoints we found a significant positive correlation between the percentage of activated CD8+ T cells (population 3, CD8+ CD69+ T cell producing IFNγ and TNF) and blood glucose levels ($r = 0.41$, $p = 0.0003$; Fig. 5o–p). Plasma GDNF levels were also significantly correlated with the number of activated CD8+ T cells

producing TNF and IL-2 (population 4; Fig. 5o, q). These data show heightened cytokine production by CD8+ T cell in response to polyclonal stimulation correlates with hyperglycemia in SARS-CoV-2-infected AGMs.

## Analysis of SARS-CoV-2 persistence in tissues 18 weeks post infection

Data suggest extrapulmonary presence of SARS-CoV-2 RNA in human tissues post-acute infection[43], and replicating virus has been isolated

**Fig. 4 | Plasma levels of GDNF and CCL25 correlate with blood glucose in unvaccinated animals (*n* = 9). a** A correlation matrix depicting the *r* values between SARS-CoV-2-modulated analytes at week 1, and serum glucose concentrations at various timepoints. Blue-colored correlations = positive correlations and red-colored correlations = negative correlations, dot size denotes size of absolute *r* value (raw *p* value correlation matrix in Fig. S4). Spearman's rank correlation was used for statistical analysis. **b** Two-sided Spearman's correlations between plasma CCL19 or GDNF with glucose across baseline, week 1, 4, and 12 in the unvaccinated group, *n* = 9 animals. **c** GDNF levels at BL and week 1, 4, 12 in unvaccinated animals (*n* = 9) p.i. Statistical significance was determined using the two-sided Wilcoxon matched-pairs signed-rank test. **d** Two-sided Spearman's correlations between plasma CCL19 and GDNF across baseline, week 1, 4, and 12 in the unvaccinated group. **e** Comparative analysis showing plasma GDNF levels in unvaccinated (*n* = 10) and vaccinated (*n* = 5) animals at baseline (BL) and weeks 1–3. Error bars represent mean with SD. Statistical comparison performed using two-sided Wilcoxon matched pairs signed-rank test. **f** Two-sided Spearman correlation between blood glucose and insulin across baseline, week 1, 4, 8, 12, and 16 p.i. in the unvaccinated group. ELISA for insulin was conducted in triplicates for each animal. **g** Unbiased STRING analysis, using custom setting with GDNF as the sole input. **h** BioGRID protein-protein analysis tool using customs settings and *Homo sapiens* database used to validate GDNF interactions (proteins of interest highlighted by arrows). *n* denotes number of independent samples/animals at each timepoint. Source data are provided as a Source Data file.

from human hepatocytes from postmortem COVID-19 patients[21]. We therefore took advantage of RNAscope using an anti-sense probe targeting SARS-CoV-2 spike protein RNA (SARS-S) but observed no substantial percentage of cells containing spike RNA (SARS-S+ cells %) in duodenum (Fig. 6a, d, g; mean = 0.018%), liver (Fig. 6b, e, h; mean = 0.005%), and pancreas (Fig. 6c, f, i; mean = 0.012%) at 18 weeks p.i., as well as in historical sections (4 weeks p.i.; mean: duodenum = 0.0037%, liver = 0.0016%, pancreas = 0.004%). Substantial SARS-S signals from lung samples of a SARS-CoV-2-infected AGM (4 weeks p.i.; Fig. 6j, k) support reliability of staining. For confirmatory study we conducted qPCR analysis on tissues collected from 14 animals at 18 weeks p.i. We found no SARS-CoV-2 subgenomic N nor subgenomic E signals in the liver, duodenum, or pancreas, but low-levels of genomic N signals (Ct mean 30.4 and 31.6) in the duodenum of two of the 14 animals. We observed even lower genomic N signals in 3 lung samples (Ct mean 33.1, 33.5, and 34.8), and since the negative PCR control recorded a mean ct value of 36.3 we consider expression in lungs to be negligible (Fig. S6). We conducted SARS-CoV-2 immunohistochemistry on duodenum collected 18 weeks p.i. but found no positive signals (Fig. 6l, m). Together these data suggest no substantial long-term persistence of replicating virus in the liver, pancreas, or duodenum in our AGM PASC model.

### Absence of severe lung inflammation and injury during long-term follow-up in SARS-CoV-2-infected AGM
We previously reported two of four AGM (2/4) exposed to SARS-CoV-2 isolate USA-WA1/2020 progressed to acute respiratory distress syndrome (ARDS) by day 8 and 22 p.i. and exhibited diffuse alveolar damage and bronchointerstitial pneumonia[26]. We therefore examined whether viral-induced high-grade lung inflammation is present at our 18 weeks study endpoint. We conducted histopathological analysis of lungs from all study animals at necropsy (*n* = 15, endpoint 18 weeks p.i. for 14 animals), and included 3 uninfected animals, and four infected from a previous study with endpoint between 3 and 4 weeks p.i. The inflammation grades were scored 1 to 4 (minimal to severe) based on indices of inflammation, utilizing the numbers of inflammatory cells, the degree of fibrous connective tissue formation (Fig. S7a, A–D), and the presence of pneumocyte type II hyperplasia (Fig. S7b, A, B). Analysis of the right lower lung shows minimal inflammation in all except one animal in the long-COVID study group. Two of the four animals from the short-term study group had severe inflammation. We generated a composite inflammation score by pooling scores from the right anterior upper, right middle dorsal and right lower lung but found no overall severe inflammation above that of the three uninfected AGM controls (Fig. S7c). There was no evidence of aspirated pneumonia or euthanasia artifacts in the two AGMs from the short-term study, with severe lung inflammation, suggesting this may be SARS-CoV-2-related. The data suggest that, at least in the animals examined by our inflammation measurements, there was no general severe lung inflammation at 18 weeks p.i.

### Absence of severe pancreatic inflammation and injury during long-term follow-up in SARS-CoV-2-infected AGM
Extrapulmonary manifestations of SARS-CoV-2 infection includes infection of the human exocrine and endocrine pancreas, causing morphological changes that may contribute to impaired glucose homeostasis[27,28,35]. We examined the inflammation and fibrosis status of hematoxylin and eosin (H&E) stained pancreatic cross sections, obtained at necropsy, and found none to minimal inflammation, like the two uninfected historic samples. There was no evidence of pancreatic fibrosis in 13 animals, while 2 demonstrated mild fibrosis. Representative images are shown (Fig. S8). Taken together, we saw no evidence to support significant long-term morphological defects of the pancreas.

### SARS-CoV-2 infection is associated with increased liver glycogen levels
Although previous reports show SARS-CoV-2 infection of pancreatic β-cells in humans and AGMs, infection in humans may be independently associated with hyperglycemia regardless of β-cell function[21]. SARS-CoV-2-infected hepatocytes have raised glucose production through increased gluconeogenesis, a potential cause of hyperglycemia in infected humans[21]. Regardless of the underlying source, elevated blood glucose and hepatic glucose uptake may provide substrate for hepatic glycogen synthesis[44] (Fig. 7a). We therefore quantified liver glycogen in uninfected and 4-weeks p.i. SARS-CoV-2 (short-term infection) infected historical samples, as well as at necropsy (long-term infection) using Periodic acid Schiff (PAS) staining. Representative staining is shown without diastase (Fig. 7b, upper panel), and with diastase (Fig. 7b, lower panel) to confirm staining specificity. A trend for increased liver glycogen in infected animals was observed (Fig. 7c), reaching significance in the longer-term infected unvaccinated group. Liver glycogen levels correlated positively with blood glucose levels at week 8 (Fig. 7d) and 12 (Fig. 7e, f) p.i. Of note, we saw no evidence of hepatic steatosis or fibrosis at necropsy.

## Discussion
Collateral damage by early host antiviral responses against SARS-CoV-2 may underlie the severity and clinical presentation of acute COVID-19 symptoms. Pre-existing or virus-induced metabolic provocations may promote or exacerbate these acute symptoms, as well as contribute to the development and long-term phenotype of metabolic syndromes such as MAFLD[14], and hyperglycemia[37]. Here we describe the first study to systemically evaluate, in NHPs, the temporal changes of hyperglycemia (metabolic PASC) over an 18-week period. We observed, in SARS-CoV-2-infected AGMs, a multitude of immunologic and metabolic changes that reflect those previously reported in humans in the acute and post-acute phase of COVID-19.

Our data demonstrate that SARS-CoV-2 infection of AGMs is associated with early-onset hyperglycemia, which persisted for at least 18 weeks p.i. We have previously reported that early immune changes contribute to adverse pathological events in SARS-CoV-2-infected NHPs[45]. Here, we identified a set of plasma analytes, many differentially

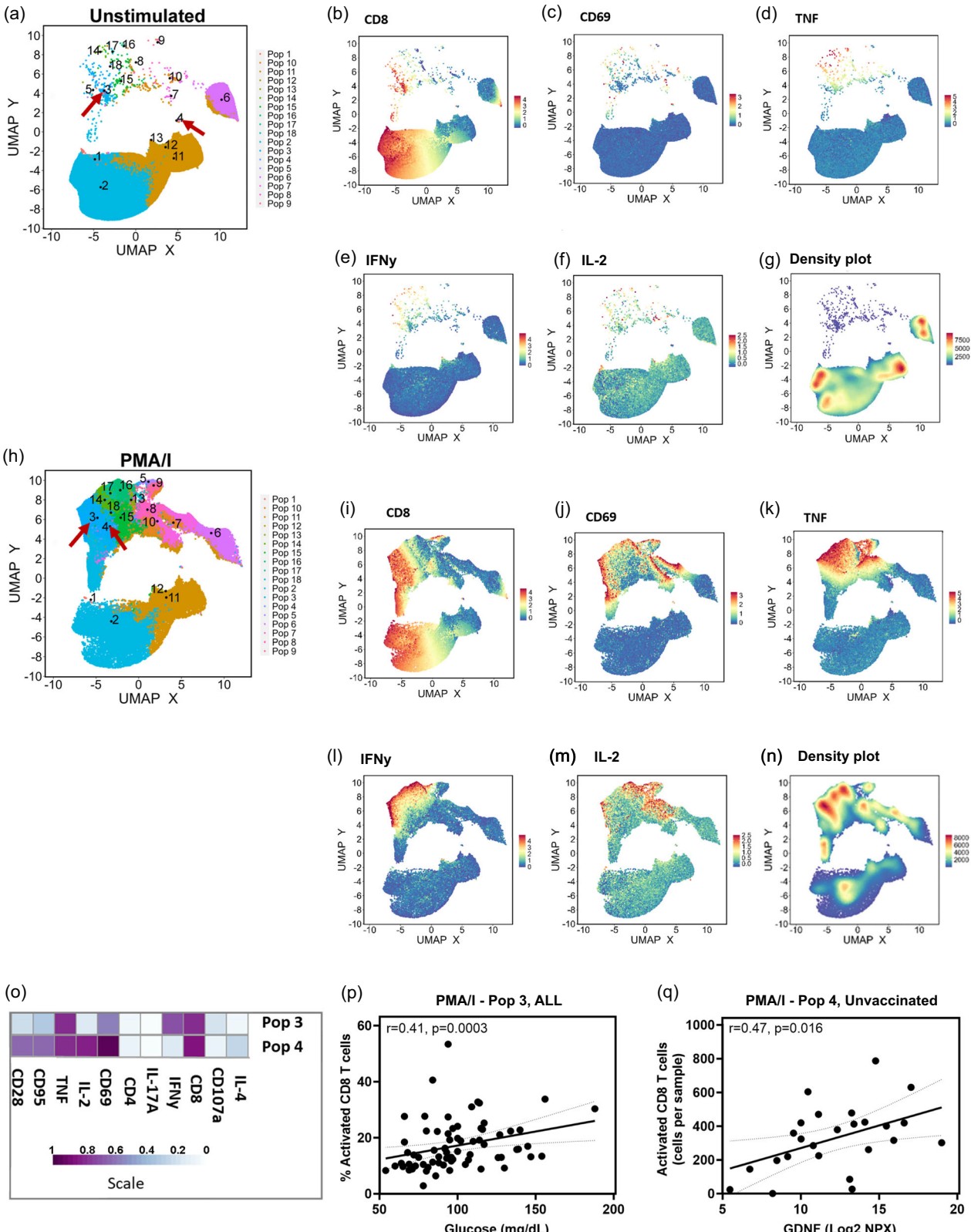

increased at week 1 p.i., that correlated positively and significantly with serum glucose levels over time. The signature of these regulated analytes was defined by a preponderance of chemokines and several inflammatory-related proteins. Using protein-protein interaction tools and gene-ontology analyses, we have discovered that the top functionally enriched networks associated with these analytes are related to leukocyte migration, chemotaxis, macrophage proliferation, and

viral protein interaction with cytokines. Amongst these analytes, we noted CCL25 and GDNF to be significantly elevated at week 1 p.i. and beyond and correlated positively with blood glucose levels across all timepoints evaluated. Thus, using the highly specific PEA technology, we identified a set of differentially regulated plasma proteins associated with early and persistent impairment of glucose homeostasis in SARS-CoV-2-infected AGMs. Additionally, we report increased

**Fig. 5 | Multi-level analysis by Spectre shows the magnitude of polyclonal responsive activated CD8 T cell populations correlating with serum glucose and GDNF levels. a** UMAP showing 18 subpopulations within CD3+ T cells in unstimulated PBMCs. Expression of selected markers is shown (**b**) CD8 (**c**) CD69 (activation), and (**d**–**f**) inflammatory markers in unstimulated PBMCs. **g** UMAP density plot of unstimulated cells. **h** UMAP showing 18 subpopulations within CD3+ T cells in PMA/I stimulated PBMCs. Expression of selected markers are shown **i** CD8, **j** CD69 (activation) and (**k**–**m**) inflammatory markers in unstimulated PBMCs.

**n** UMAP density plot of PMA/I stimulated cells. **o** Heat maps showing the defining phenotypes of population 3 and 4. UMAPS are the composite of 10 unvaccinated AGMs. **p** Two-sided Spearman correlations between blood glucose levels in all animals and percentage of population 3 within CD3+ T cells in the PMA/I treatment. Correlation conducted between glucose and cell populations at BL and weeks 1, 4, 8, and 12. **q** Two-sided Spearman correlations between plasma GDNF levels in unvaccinated animals and a number of activated CD8+ T cells within CD3+ T cells in the PMA/I treatment. Source data are provided as a Source Data file.

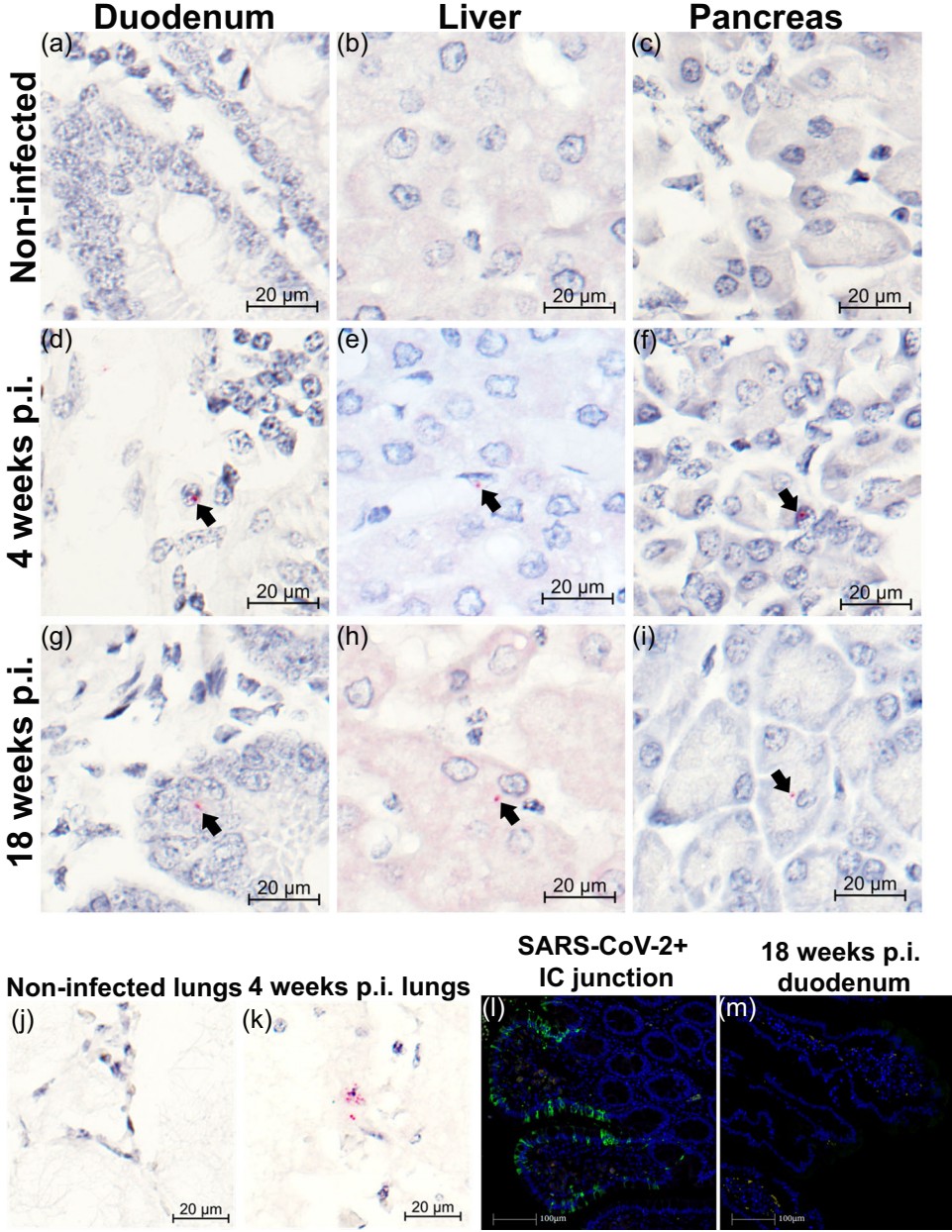

**Fig. 6 | Evaluating the persistence of SARS-CoV-2 RNA in tissues of AGMs up to 18 weeks post infection. a–c** Representative images of RNAscope RED using the SARS-Spike (S) probe in duodenum, liver, and pancreas in non-infected animals (*n* = 2), (**d**–**f**) 4 weeks post infection (p.i., *n* = 4), and (**g**–**i**) necropsy/18 weeks p.i. (*n* = 15). RNAscope RED was used to visualize the SARS-S expression frequency in the tissues, counterstained with hematoxylin. Black arrows note the presence of SARS-S copies as red dots by colorimetric RNAscope. **j**–**k** Representative validation images of RNAscope RED using the SARS-S probe in lungs of non-infected, and 4 weeks p.i. animals. Tissues were counterstained with hematoxylin. **l**, **m** Fluorescent immunohistochemistry in a representative SARS-CoV-2 positive control from 5 technical replicates (**l**) and a representative stain from 14 animals 18 weeks p.i. (**m**). Blue = DAPI; Green = anti-SARS antibody. Staining was performed once in each tissue in tandem with controls of the same tissue. *n* denotes number of independent animals per group.

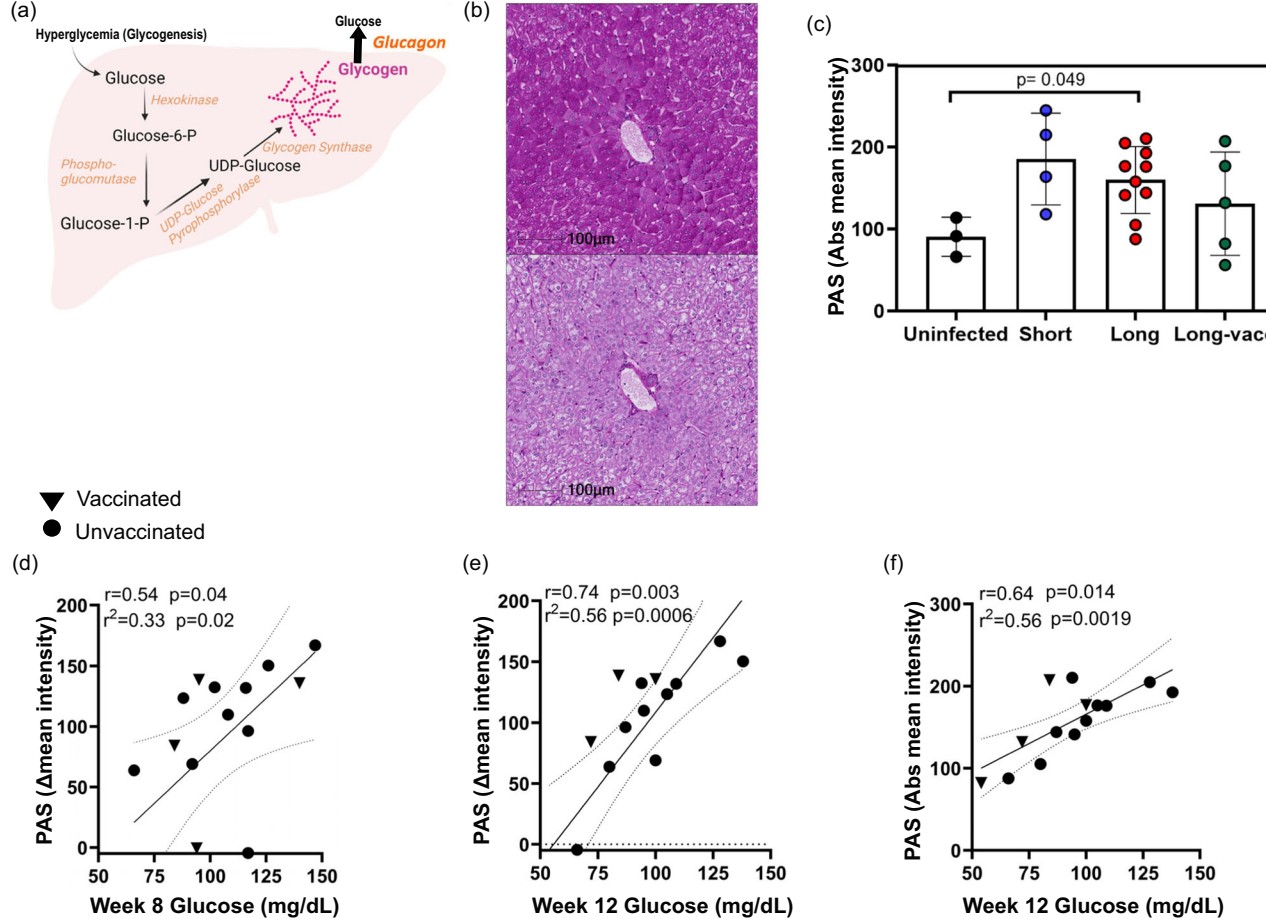

**Fig. 7 | Liver glycogen levels and relationship with blood glucose. a** A model of liver glucose homeostasis during SARS-CoV-2 infection. Elevated blood glucose triggers glycogenolysis, which stores glucose into glycogen. Increased glucagon levels in SARS-CoV-2 infection stimulates the conversion of stored liver glycogen back to glucose perpetuating a vicious cycle. Figure 7a Created with BioRender.com released under a Creative Commons Attribution-NonCommercial-NoDerivs 4.0 International license. **b** Representative image of Periodic acid Schiff staining of AGM liver at 18 weeks post infection (p.i.) without diastase (upper panel), and with diastase (lower panel). **c** Glycogen levels in livers from uninfected AGMs ($n = 3$), 4-week p.i. (short-term, $n = 4$) and at necropsy/18-week p.i. (long-term unvaccinated, $n = 10$ or vaccinated, $n = 5$). Statistical comparison between groups was done using the two-sided Mann–Whitney $U$ test. Error bars represent mean with SD. **d**–**f** Two-sided Spearman's correlation between blood glucose at various weeks and hepatic glycogen at 18 weeks p.i. in unvaccinated (circles) or vaccinated (triangles). Intensity of stain was quantified as either delta mean intensity (difference between diastase treatment and untreated), or absolute intensity. $n$ denotes number of independent samples/animals at each timepoint. Source data are provided as a Source Data file.

cytokine production by T cells ex-vivo in response to PMA/I stimulation positively correlating with glucose levels. Finally, while the magnitude of antibody responses was similar in the vaccinated and unvaccinated groups, we show that the vaccination of five animals 4 days p.i. was associated with a consistent and significantly lower blood glucose level over the study period.

Previous reports show that elevated glucose levels favor SARS-CoV-2 infection and monocyte proinflammatory responses[46], as well as elevating the risks of severe COVID-19 progression and increased fatality[18,19]. In addition, increased cytokine production in response to in vitro polyclonal stimulation (PMA/I) is seen in severe and extreme cases of COVID-19[13]. Hyperglycemia may be caused by insulin resistance, dysfunctional pancreatic β-cells, impaired glucose clearance or increased glucose disposal by the liver through gluconeogenesis or glycogenolysis[21]. We have previously shown in a short-term 4-week study, in two AGMs with demonstrable SARS-CoV-2 infection of the pancreatic ductal and endothelial cells, that infection was associated with pancreatic thrombofibrosis and new-onset diabetes[35]. The focus of this AGM PASC study precludes dissecting the specific mechanisms by which glucose homeostasis is regulated during infection. Nonetheless, pathological elevation of blood glucose in moderate and severe COVID-19 patients has been linked to enhanced hepatic gluconeogenesis by the activity of the Golgi protein GP73, which is found to be elevated in the plasma of infected patients[37]. Although it remains controversial whether SARS-CoV-2 can replicate in hepatocytes, data show that these cells express the chaperone glucose-regulated protein 78 (GRP78), a putative SARS-CoV-2 entry factor, as well as low levels of ACE2 protein, which co-localized with spike protein. Moreover, the co-localization of spike protein with viral RNA, and ex-vivo infection assays support the inference that SARS-CoV-2 can replicate in the liver[21]. Although the infection appears non-cytopathic, it was proposed that SARS-CoV-2, including the major gamma, delta, and omicron variants, can stimulate hepatic glucose production and disposal through increased activity of the rate-limiting gluconeogenic enzyme phosphoenolpyruvate carboxykinase (PEPCK)[21]. The key focus of this work was to ascertain whether AGMs infected with SARS-CoV-2 are feasible models to study metabolic PACS, so detailed mechanistic analyses are beyond this study. However, we observed elevated glycogen levels in hepatocytes of infected AGMs at necropsy, which positively correlated with blood glucose levels. On the contrary we saw no substantial amount of SARS-CoV-2 nucleic acid or proteins in the liver or pancreas 18 weeks p.i. Glucagon, which can promote the

breakdown of stored hepatic glycogen to glucose, is increased in COVID-19 patients' plasma[21], but we have not interrogated this connection in our model. Nonetheless this supports the idea that non-insulin, and non-pancreatic related mechanisms may partially regulate glucometabolic control during SARS-CoV-2 infection[37]. While the persistence of viral nucleic acid and antigens in some extrapulmonary tissues have been linked to PASC in patients with underlying illnesses[47], questions remain on how clinically relevant these "viral ghosts" are, and the mechanisms by which they contribute to PASC. In fact, quantification of SARS-CoV-2 RNA (N) using the highly sensitive ddPCR in autopsy liver and pancreas from 11 late case patients who died with COVID-19, yielded negative in most of the deceased samples[43]. In the same study, a significant proportion of deceased patients had undetectable or low viral RNA signals (<0.1 N gene copies per nanogram input RNA) in the intestine[43]. We detected no significant levels of viral RNA in lungs, but none of our study animals died of COVID-19 complications. Hence, the higher viral burden detected in respiratory sites in some deceased COVID-19 patients may reflect a link between viral load and mortality[43].

To gain insights into other potential processes driving COVID-19-related hyperglycemia, we assessed a panel of blood inflammatory/metabolic analytes at baseline and at set study timepoints. The signature of proteins differentially increased at week 1, and associated with serum glucose levels over time, was biased towards those involved in leukocyte migration and chemokine receptor binding. Some of these include CCL8, CCL19, and the gut-homing chemokine CCL25, which correlated strongly and positively with blood glucose levels across multiple weeks.

Recent studies show plasma chemokines to be critical factors that control COVID-19 severity[48,49]. Accordingly, CCL25 has been found to be elevated in the plasma of COVID-19 patients[50], and a large-scale genome wide association study, involving patients with severe COVID-19, identified mutations in several chemokine receptors, including the CCL25 receptor CCR9 as a major risk factor for developing severe COVID-19[51]. We found no evidence of long-term pancreatic damage, however the elevation of CCL25 could potentially impair insulin secretion from the pancreas.

Intriguingly, while anti-SARS-CoV-2 antibodies may counter viral replication, antibodies against inflammatory mediators such as CCL25, may be more effective at counteracting development of both acute and long-COVID. We found our set of differentially regulated chemokines (CCL8, CCL19, and CCL25) intriguing because autoantibodies against CCL8 are augmented in long-term convalescent COVID-19 individuals, and elevated antibodies against the COVID-19 signature chemokine CCL19[49] are documented with high confidence in both acute and long-term COVID-19 phases compared to uninfected controls[52]. Interestingly, autoantibodies against CCL25 are augmented in mild COVID-19 patients compared to those requiring hospitalization[52]. Convalescents exhibiting PASC at 12 months have a significantly lower cumulative level of anti-chemokine antibodies at six months compared to those who reported no PASC[52]. In our STRING protein-protein analysis of significantly regulated analytes at week 1, we found strong interactions between CCL8, CCL19, CCL25, TNF, and IL-18, confirming a link between this chemokine signature and inflammatory responses in our infected AGM model, likely contributing to early adverse and long-term pathological events.

Some evidence suggests a link between inflammatory processes and COVID-19-related metabolic diseases, but our understanding of the underlying mechanisms remains limited. Chemokines are best known for their role in immune cell trafficking to sites of infection and as mediators of inflammation and tissue repair. However, recent reports have linked their activity to features of metabolic syndrome such as insulin resistance and T2D. CCL25 acting via its receptor CCR9 impairs β-cell function and inhibits glucose-induced insulin secretion[30]. Moreover, CCR9 has been implicated in the pathogenesis of T2D by

modulating small intestine permeability and inflammation[53]. Although we found a strong and significant positive relationship between blood glucose and CCL25 levels, linking this causatively to glucose homeostasis in COVID-19 requires further investigation.

Besides the classic chemokines, we discovered GDNF to positively associate with serum glucose levels over time. GDNF is a neurotrophic factor belonging to the transforming growth factor-β (TGF-β) superfamily, which plays a key role in the nervous system, and the pathogenesis of mood disorders. GDNF is a known canonical RET ligand, validated by our STRING analysis, which demonstrates a high-confidence interaction between GDNF and RET. While the role of GDNF in peripheral glucose metabolism is unclear, another RET ligand, GDF15, also belonging to the TGF-β superfamily, is known to regulate systemic metabolic homeostasis and is a correlative biomarker for metabolic syndrome[54,55]. Moreover, GDF15 binds with high affinity to GDNF family receptor α-like (GFRAL), an interaction required for GDF15-RET binding[56,57] and may represent a compensatory checkpoint during conditions of high metabolic stress such as SARS-CoV-2 infection. In fact, GDF15 levels are elevated in SARS-CoV-2-infected patients and are significantly associated with worse clinical outcomes[58].

GDNF has been shown to reverse the pathological effects of hyperglycemia on enteric neuronal survival via activation of the PI3K-Akt pathway[59], a signaling cascade that regulates Glut1 and Glut4 mediated glucose uptake into cells[60,61]. Recently, nutritional regulation of GDNF has been suggested, in which its expression was enhanced by glucose[62]. Interestingly, in protein-protein interaction analysis, besides its receptors, GDNF also interacts with Neural Cell Adhesion Molecule 1 (NCAM1) with high confidence, confirming its potential role as a chemotaxis factor, especially for epithelial and enteric neural cells essential for maintaining gut wall integrity[63,64]. GDNF levels correlate inversely with plasma glucose in T2D patients[65], and have been shown to improve glucose tolerance and increase β-cell mass in vitro and in vivo[66]. Thus, the increased GDNF in our studies is likely an adaptive response. The precise role of GDNF in COVID-19 related pathologies is unknown but may represent a compensatory immunometabolic adaptation related to changes in energy metabolism in infected AGMs.

In conclusion, we show SARS-CoV-2-infected AGMs exhibit many virologic, immunologic, and metabolic features observed in infected humans and may represent a useful model to interrogate early, and persistent factors associated with metabolic PASC. We identify GDNF and several plasma analytes, dominated by chemokines, that are associated with hyperglycemia over several months p.i. We provide leads involving inflammatory processes, as well as potentially dysregulated liver glucose homeostasis, that warrant further investigation to improve our understanding of how early inflammatory and metabolic responses against SARS-CoV-2 infection influence its severity and long-term metabolic complications. Such understanding may provide the basis for exploring autoantibodies of chemokines/metabolic-regulating factors to treat and prevent long-COVID.

In a large-scale PASC study in humans utilizing the Olink Explore 384 Inflammation panel, the pattern of regulated plasma proteins did not overlap with our results. The PASC evaluated were related to fatigue, anxiety/depression, gastrointestinal and cardiorespiratory symptoms, and not to classic metabolic diseases (e.g., hyperglycemia/diabetes). Indeed, the authors found that although persistent inflammation defined PASC, distinct profiles were associated with specific disease manifestations[67].

Although SARS-CoV-2 infection is linked to a higher risk of diabetes in the general population, a limitation of our study is the bias towards female AGMs. Future work should examine any potential impact of sex on mechanisms that link infection to metabolic PACS. Another limitation is that not all data from the OLINK analysis met our QC standards, potentially influenced by antibody-reactivity issues for some proteins. We conducted STRING pathway analysis on these

"failed" proteins and found them to be significantly related (FDR < 0.05) to interleukin signaling based on Local Network Cluster-STRING, Wiki Reactone, and Kegg pathways. Moreover, since interferons were under-represented in the OLINK inflammation panel, we cannot exclude the possibility that these processes do not contribute to persisting hyperglycemia post-acute SARS-CoV-2 infection.

Since mRNA vaccines may elicit an immune response within hours and induce humoral immunity within 5 days of administration[68], it is plausible that such responses may offer favorable immunometabolic benefits prior to multiorgan distribution of SARS-CoV-2 produced by viral shedding from the lungs into body fluids. Intriguingly, in an observational cohort study of 15 million people COVID-19 vaccination reduced the incidence of long-term diabetes significantly[69]. Our observation of better glycemic control in the vaccinated group requires further studies to evaluate the potential benefits of vaccination during the acute phase of infection[70].

## Methods

### Study approval
This study was reviewed and approved by the Institutional Animal Care and Use Committee of Tulane University. Animals were cared for in accordance with the NIH's *Guide for the Care and Use of Laboratory Animals*. Procedures for handling and BSL2, and BSL3 containment of animals were approved by the Tulane University Institutional Biosafety Committee. The Tulane National Primate Research Center is fully accredited by the Association for Assessment and Accreditation of Laboratory Animal Care.

### Animals and infection procedure
Procedures are in accordance with those we have previously reported[26]. Briefly, we exposed 15 African green monkeys (*Chlorocebus aethiops sabaeus*; 13 females, 2 males) aged 7.92 to 19.32 years, to SARS-CoV-2 strain 2019-nCoV/USA-WA1/2020 at ~1e6 TCID50 via intranasal (0.5 mL/nares), and intratracheal (1 mL) routes. Except for one (PB24), obtained from the NIH via the Wake Forest breeding colony, all animals were of Caribbean origin (wild-caught) purchased from Bioqual (MD, USA). Ten animals (9 females) were studied during the natural course of SARS-CoV-2 infection and 5 animals (4 females) received the BNT162b2 Pfizer/BioNTech vaccine 4-days post infection. Animals were monitored daily for 18 weeks. Animals were anesthetized with telazol tiletamine hydrochloride and zolazepam hydrochloride (5 to 8 mg/kg intramuscular; Tiletamine–zolazepam, Zoetis, Kalamazoo, MI) and buprenorphine hydrochloride (0.03 mg/kg).

There is an overabundance of females in the study based on the limited availability of animals that were assigned to this study. A skewed sex ratio is sometimes the case with NHPs as they are a limited resource. This is especially true for AGMs, which are not bread at the TNPRC, and have been imported from wild-caught animals.

### Blood chemistry and hematological analysis
A comprehensive biochemistry analysis on blood EDTA-collected serum was performed at the TNPRC clinical lab, using the Beckman Coulter AU480, according to the manufacturer's instructions. The panel included albumin, glucose, cholesterol, triglycerides, aspartate aminotransferase (AST), alanine aminotransferase (ALT), blood urea nitrogen (BUN), alkaline phosphatase (ALP), and lactate dehydrogenase (LDH). Hematological analysis on whole blood, including absolute quantification, and percentages of neutrophils, monocytes, lymphocytes, and eosinophils were performed on the Sysmex NX-V-1000 Hematology Analyzer.

### Virological analysis: Genomic and subgenomic RNA quantitation
Pre- and post-exposure samples of mucosal swabs (nasal and pharyngeal brush) were obtained for virological analysis. For RNA extraction from swab samples 200 μL of 1× DNA/RNA Shield (Zymo; Cat no. R1100) was added to each swab and RNA was extracted using the Zymo Quick RNA Viral Kit (Zymo; Cat no. R1035) according to manufacturer's instructions. Samples were eluted in 50 μL volume. Subgenomic and genomic SARS-CoV-2 mRNA were quantified as previously described using appropriate primers/probes, and cycling conditions[71,72], with the exception that, the probe for detection of the subgenomic nucleocapsid RNA was modified as described[73]. Briefly, qPCR analysis was conducted on a QuantStudio 6 (Thermo Scientific, USA) using TaqPath master mix (Thermo Scientific; Cat no. A15299). Signals were compared to a standard curve generated using in vitro transcribed RNA of each sequence diluted from $10^8$ down to 10 copies. Positive controls consisted of SARS-CoV-2-infected VeroE6 cell lysate. Viral copies per swab were calculated by multiplying mean copies per well by volume of swab extract. The following primers were used, genomic N; forward 5′-GAC CCC AAA ATC AGC GAA AT-3′, reverse 5′-TCT GGT TAC TGC CAG TTG AAT CTG-3′, subgenomic N; forward 5′-CGATCTCTTGTAGATCTGTTCTC-3′, reverse 5′-GGTGAACCAA-GACGCAGTAT-3′, and subgenomic E; forward 5′-CGA TCT CTT GTA GAT CTG TTC TC-3′, reverse 5′-T GTG TGC GTA CTG CTG CAA TAT-3′.

### Vaccination
10 animals were studied during the natural course of SARS-CoV-2 infection, and 5 animals received one dose of the BNT162b2 (Pfizer/BioNTech) vaccine 4 days post-infection.

### Antibody response analysis
Detection of anti-SARS-CoV-2 antibodies against spike (S), spike S1RBD (S1RBD), and nucleocapsid (N) were performed using MSD S-PLEX CoV-2, MSD S-PLEX CoV-2 S1RBD and MSD S-PLEX CoV-2 N assay kits for IgA, IgM, and IgG antibodies (Meso Scale Discovery, Rockville, MD). The assays were done according to the manufacturer's instructions. Plasma samples were diluted 1/500-fold in the assay buffer provided. The plates were read using the MESO QuickPlex SQ 120MM reader. Sample quantitation was achieved using a calibration curve generated using a recombinant antigen standard. During analysis, any concentrations below the limit of detection (LOD) were assigned the LOD value, and any concentrations above the highest calibration standard were assigned its value.

### In vitro polyclonal stimulation
Cell Stimulation Cocktail (eBioscience) containing phorbol 12-myristate 13-acetate (PMA), ionomycin, brefeldin A and monensin were used ×1 to stimulate PBMCs. Briefly, PBMCs were thawed, stimulated, and incubated for 6 h in supplemented RPMI-1640 medium [10% human serum, penicillin/streptomycin (Invitrogen), 2 mmol/l L-glutamine (Invitrogen, Carlsbad, California, USA)] at 37 °C, 5% $CO_2$. PBMCs were stained with Zombie Aqua (BioLegend, Cat no. 423101) for live/dead cell gating, and surface stained using the following pre-titrated antibodies: CD45-PerCP (BD, clone D058-1283, Cat no. 558411), CD3- BV650 (BD, clone SP34-2, Cat no. 563916), CD4- BV786 (BD, clone L200, Cat no. 563914), CD8-BUV737 (BD, clone RPA-T8, Cat no. 749367), CD28- BV605 (BD, clone CD28.2, Cat no. 562976), CD95- BV711 (BD, clone DX2, Cat no. 563132), CD69- PE-CF594 (BD, clone FN50, Cat no. 562617) and CD107a- BUV395 (BD, clone H4A3, Cat no. 565113). Except for CD45-PerCP (NHP), all antibodies have verified reactivity to humans and cross-reactivity with NHPs. Cells were fixed and permeabilized using BD Fixation/Permeabilization solution (BD Biosciences) and stained for intracellular cytokines using the following antibodies from BD Biosciences: IL-2- BB700 (BD, clone MQ1-17H12, Cat no. 566405), IL-4- BV421 (BD, clone 8D4-8, Cat no. 562986), TNF- APC (BioLegend, clone MAb1, Cat no. 502912), IFNy- PE-Cy7 (BioLegend, clone 4 S.B3, Cat no. 502528), IL-17A- PE (BioLegend, clone BL168, Cat no. 512306). Information on antibody validation and concentration is

found in Table S4. Cells were fixed in 2% PFA and acquired on a BD FACSymphony™ by the Flow cytometry core (TNPRC). Data were analyzed using FlowJo software (v 10.8.1, Tree Star Inc., Ashland, Oregon, USA) or used for Spectre analysis in R (v4.2.1)[74,75].

## Plasma analyte (OLINK) analysis
Plasma analytes were analyzed using a PEA (Olink, Proteomics)[76]. Plasma was collected from freshly collected blood in EDTA anticoagulant tubes and centrifuged at 650 × g for 10 minutes. The plasma was aliquoted to minimize freeze-thawing and stored at −80 °C. Samples were processed at the OLINK Analysis Services Lab in Waltham (MA, USA) or within the High Containment Research Performance Core (TNPRC). The Olink® Target 96 inflammation panel (Olink Proteomics AB, Uppsala, Sweden) was used to measure proteins following manufacturer's instructions. In brief, pairs of oligonucleotide-labeled antibody probes are mixed with plasma to allow binding to their targeted protein. Oligonucleotides will hybridize in a pair-wise manner if the two probes are brought in proximity. A reaction mixture containing DNA polymerase allows proximity-dependent DNA polymerization and creates a unique PCR target sequence. The amplified DNA sequence is quantified, quality-controlled, and normalized using internal control and calibrators. Technical replications were done in duplicates. Protein levels are expressed as arbitrary units NPX values.

## Insulin determination
Insulin plasma levels were measured using the monkey insulin ELISA kit (AssayGenie, Dublin, Ireland). The essay was conducted according to manufacturer's instructions using 1:8 diluted samples.

## RNAScope analysis
Formalin-fixed paraffin-embedded (FFPE) tissues were collected at necropsy and sectioned at 5 µm. In-situ hybridization (ISH) was conducted using RNAscope® 2.5 High Definition (HD) RED Assay Kit (Advanced Cell Diagnostics), according to the manufacturer's directions. Briefly, FFPE tissue sections were deparaffinized in xylenes and dried, followed by incubation with hydrogen peroxide. Heat-mediated antigen retrieval was carried out in a steamer with the provided kit buffer. Samples were treated with the kit-provided protease and hybridized with the V-nCoV2019-S probe in a HybEZ oven (Advanced Cell Diagnostics). All washes were performed with the kit wash buffer. Signal amplification was accomplished with six successive AMP solutions and the kit-provided Fast Red dye. Slides were counterstained with hematoxylin. Control slides were included in every run to confirm the specificity of staining and assess the background.

For imaging and quantitation, brightfield images were acquired using the Axio Observer 7 (Zeiss), equipped with ZEN blue edition software (v3.6.096.08000). Images were subjected to brightness and contrast enhancement in Photoshop (Adobe, v24.4.0) applied to the entire image. Slides were scanned with the Axio Scan.Z1 digital slide scanner (Zeiss). Scanned files were analyzed with HALO (Indica Labs, v3.4.2986.151) algorithm ISH (v4.1.3) for a non-biased measurement of copies on a cell-by-cell basis. The ISH algorithm was run in annotations specific to the tissue section of interest, using hematoxylin-stained nuclei to quantify the number of cells, and Fast Red intensity and size accounted for the positivity of the probe within the cell. Each resulting count was assessed individually, and all false positives were excluded.

## SARS-CoV-2 immunohistochemistry
Formalin-fixed, paraffin-embedded tissue sections were deparaffined using standard procedures followed by heat (microwave) induced antigen retrieval in a high pH solution (Vector Labs H-3301), rinsed in hot water and placed in heated low pH solution (Vector Labs H-3300) and allowed to cool to room temperature. Sections were washed in phosphate-buffered saline, blocked with 10% normal goat serum (NGS) for 40 minutes, and incubated for 60 minutes with a 1:1000 dilution of guinea pig anti-SARS antibody (BEI NR-10361). The slides were incubated with a 1:1000 diluted goat anti-guinea pig secondary Alexa Fluor 488 conjugated antibody (Invitrogen, A11073) for 40 minutes. Nuclei were labeled with DAPI (4′,6-diamidino-2-phenylindole). Images were taken with a Zeiss Axio Scan.Z1 slide scanner and analyzed using HALO HighPlex FL v4.1.3 (Indica Labs).

## Periodic acid schiff
Slides were deparaffinized on an auto Stainer (Histology Core, TNRPC) and subjected to periodic acid schiff hematoxylin stain for glycogen. Briefly, slides were either treated with a 1% solution of diastase (control) or not, and samples oxidized with 1% Periodic Acid (Poly Scientific) for 10 minutes, washed, placed in Schiff's reagent for 10 minutes, counterstained with hematoxylin and eosin (H&E), and coverslipped using standard procedures. Stain intensities were quantified using automated settings in ImageJ1.53t (Fiji)[77].

## Histopathological scoring
All relevant tissue samples were fixed, paraffin-embedded and stained with H&E for histopathological analysis of inflammation, fibrosis, or other relevant pathologies by an experienced, board-certified pathologist.

## Bioinformatics (protein-protein interactions & pathway analysis)
Protein-protein interaction analysis was performed mainly using STRING (Search Tool for the Retrieval of Interacting Genes/Proteins (v 11.5)[78]. The *Homo sapiens* database was selected for the input data search. Unless otherwise noted, the default settings were used, including 10 as the maximum number of interactors to show, and minimum required interaction score was set to medium confidence (0.400). The top 3 functional enriched networks with FDR < 0.05 are reported. Functional classification of proteins was determined using PANTHER (Protein Analysis Through Evolutionary Relationships; v 17.0; http://www.pantherdb.org/) classification system[79,80]. BioGRID (v4.4; https://thebiogrid.org/), a biomedical interaction repository, was also used to validate interactions between entries.

## Spectre analysis
Spectre analysis was conducted as previously published[74]. The "CSV−Scale values" for each cell population of interest were exported from FlowJo. The exported data were analyzed using the Spectre (v 1.1.0) R package workflow. Briefly, the flow cytometry standard data were arcsinh transformed and clustered with FlowSOM. The clustered data were down-sampled, and dimensionality reduction was performed with UMAP. The clusters were manually labeled as desired. Finally, the clusters were used to generate summary statistics, which were used for further statistical analysis.

## Statistical analysis and packages
R was used to perform statistical analysis, principal components analysis, and ggplot2 (v3.3.3) was used to create principal component plot, heat maps, and correlation matrices. To visualize the longitudinal changes in antibody levels over time, the data were log10-transformed. We employed the LOESS (Locally Estimated Scatterplot Smoothing) regression method of the ggplot2 package for its ability to create smooth curves that effectively capture underlying trends and variations in the data. By utilizing ggplot2 in R, we enhanced scatterplots with these smooth LOESS regression curves, providing a clear representation of the evolving trends in both variables over time.

Additional graphs were created using GraphPad Prism (v 9.0). Wilcoxon matched pairs signed-rank test was used for paired analysis, Spearman's rank correlation for correlation, Mann−Whitney *U* test for

unpaired analysis, and PERMANOVA (in R; package vegan) for non-parametric ANOVA with permutations. Statistical significance is indicated by $p < 0.05$. All statistical tests are two-sided.

**Reporting summary**

Further information on research design is available in the Nature Portfolio Reporting Summary linked to this article.

## Data availability

The authors declare that data supporting the results and conclusions of this study are available within the paper and its supplementary information. Source data are provided in this paper. Additional data that support the findings of this study are available from the corresponding authors upon reasonable request. The OLINK data generated in this study have been deposited in the Figshare database license CC BY 4.0, available at https://figshare.com/s/1259afc65d4fcd09b14c[81]. STRING (Search Tool for the Retrieval of Interacting Genes/Proteins (v 11.5; https://string-db.org/) and BioGRID (v4.4; https://thebiogrid.org/) were used to conduct protein-protein interaction analysis. Codes used for data analysis are found here: https://github.com/chrysperdios/Non-Human-Primate-Model-of-Long-COVID-Identifies-Immune-Associates-of-Hyperglycemia/tree/main. Source data are provided with this paper.

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

## Acknowledgements

This study was supported by the NIH/ORIP-supported base grant to the TNPRC (P51 OD011104), and Long-COVID supplements to the P51 including P51 OD011104-60S2, 61S1, 62S2. Additional support was provided by TUHS Auxiliary Endowment for Excellence at TNRPC to

C.S.P. M.A.-M. is funded by grants from the American Lung Association and the Campbell Foundation. He also receives support from the NIH through grants R01AA029859, R01NS117458, R01AI165079, and R01DK123733. P.K.D is funded by NIH through grant R01DEO32291. NIH S10 OD026800 was awarded to support the TNRPC Flow Cytometry Core Facilities. The following RRID:SCR supported core facilities utilized by this study: Clinical Pathology Core (SCR_024609), Confocal Microscopy and Molecular Pathology Core (SCR_024613), Flow Cytometry Core (SCR_024611 and SCR_008167), High Containment Research Performance Core (SCR_024612), Virus Characterization, Isolation, Production, and Sequencing Core (SCR 024679) and Pathogen Detection and Quantification Core (SCR_024614). This study was also supported by Tulane University startup funds to JR. Facilities and equipment support to the TNPRC Regional Biocontainment Laboratory were provided by UC7AI180314 and NIAID Simian Vaccine Evaluation Unit Contract 75N93020D00007/ Task Order 75N93021F00001. The authors express enormous gratitude for the invaluable contribution of their deceased colleague Angela Birnbaum who had made invaluable contribution to the design and biosafety oversight of this study.

## Author contributions

J.R., J.P.D., R.B., T.F., R.V.B., A.B., and K.R.L. designed the animal study. C.S.P. and J. R. designed a laboratory study. C.S.P., C.C., N.G., G.L., N.J.M., P.K.D., K.M.G., L.H., K. W., M.F., K.M.G., C.M., and K.B. participated in tissue acquisition and processing and performed experiments. C.S.P., C.P., A.A.S. and T.F., M.A.-M., R.T., and P.K.D. analyzed data. J.M. provided significant and substantive data analysis support. C.S.P., J.R., C.P., A.A.S., R.T., and T.F. interpreted data. C.S.P., J.R., C.P., M.A.-M., A.A.S., C.M., and T.F. provided significant intellectual input. C.S.P. wrote the manuscript first draft. J.R., C.P., J.M., J.P.D., and M.A.-M. provided critical, and substantive intellectual editing. C.S.P., C.P., T.F., A.A.S., and M.A.-M. prepared manuscript figures. C.K. provides quality assurance and data management tasks. C.P., P. K. D., N.J.M., C.C., R.T., and T.F. edited the manuscript. All authors approved the manuscript.

## Competing interests

The authors declare no competing interests.
