## [Peer Review File · Nature Communications]

Non-Human Primate Model of Long-COVID Identifies Immune Associates of HyperglycemiaREVIEWER COMMENTS

Reviewer #1 (Remarks to the Author):

The manuscript by Palmer and colleagues investigates hyperglycemia in a non-human African Green Monkey (AGM) primate model of acute, short-term, and Long COVID or Post Acute Sequelae of COVID-19 (PASC). Additionally, the effect of vaccination during the acute phase of SARS-CoV-2 infection on glucose regulation long-term. The investigated infected 15 AGMs with 1×10^6 TCID₅₀ of the SARS-CoV-2 virus 2019-nCoV/USA-WA1/2020 through the intranasal and intratracheal route. Five of the animals received one dose of the original Pfizer/BioNTech BNT162b2 vaccine 4 days post infection. In life samples were collected throughout the study time course over 18 weeks. Samples were evaluated for viral load, serum cytokine responses, antibody responses, T cell activation, and insulin regulation. Viral RNA was found to be retained in the tissues of infected animals even 18-weeks post infection while viral RNA shedding in the upper respiratory tract had mainly cleared by week 5. The authors found that dysregulation of chemokines and hypersensitivity of T cell responses which correlated with elevated and persistent hyperglycemia at least 4 months post SARS-CoV-2 infection. GDNF increased in the non-vaccinated animals 103 weeks post infection compared to baseline which correlated with the chemokine CCL25. Interestingly, polyfunctional CD4 T cells numbers correlated with plasma glucose and GDNF levels. Glucose regulation was stabilized in animals that received a COVID-19 vaccine. In summary the authors show a potential non-human primate model of metabolic PASC. Currently, there is a lack of preclinical models which can be used to understand the mechanisms promoting Long COVID. Additionally, there is little understood about the long-term regulation of insulin and glucose following COVID-19 and how glucose regulation may play a role in PASC. The work put forth by the authors is of interest to the field as it both puts forth a potential preclinical model of PASC and offers novel findings implicating the dysregulation of glucose months following SARS-CoV-2 infection suggesting a role for glucose regulation and chemokines in the development and maintenance of Long COVID.

1. Results – Figure of study design. Figure 1a was poorly described in the results section and in the introduction. Can the authors please add a description of the study including the number of animals used, the infectious dose, route of infection, the vaccine administered, the type and number of samples collected, and specifically when the samples were collected? It was difficult to evaluate the study's results without this information. Especially missing was the description of the samples collected. It was unclear if the antibody analysis was from mucosal or blood samples.
2. Study Design Rationale. It was not clear or described why the authors decided to vaccinate the animals 4 days post infection. This study question would have been important when vaccines were first rolled out as it was unknown if vaccination in a vulnerable person during the acute phase of SARS-CoV-2 infection could decrease disease severity. However, at this time, since so many people have pre-existing immunity through vaccination or infection, it is unclear why vaccination would have been performed after infection. Can the author's please clarify this?
3. Results – Figure clarity. Many of the figures were too small to read especially in Figures 1b, 2b, 3a, 3d, 3e, 4a, 4g, 4h, 5b, and 5d. Can the authors please increase the size of the fonts? In Figure 4a, I

can't read any of the text in yellow. Can this please be corrected? Additionally, there are 2 panels labeled Figure 2f. This was confusing. Can Figure 2 be better presented?

4. Results – Figure Labeling. The graphs often do not indicate what group the samples came from – vaccinated or unvaccinated. Can the authors please add this to the graphs to make it easier to read? For example in Figure 4.

5. Results – Supplementary Tables. The supplementary Tables have a formatting issue where there is overlapping text. Can the authors please fix this in Table S1 and Table S2?

6. Results – Additional Baseline Values. The authors are evaluating Glucose, Triglycerides, and cholesterol over 18 weeks, but there are only one or two baseline values. Can the authors provide a long-term measurement of these parameters as a comparison to the data from SARS-CoV-2 infected animals?

7. Study design. Authors should explain the sex imbalance of the animals used in the study.

Reviewer #2 (Remarks to the Author):

In this study, Palmer and colleagues characterized the pathogenesis of SARS-CoV-2-infection in African green monkeys (AGMs), mostly focusing in the systemic impact of glucose metabolism and inflammation. They reported that this non-human primate (NHP) species can be used as an animal model to assess the impact of Long-Covid and to test therapeutic candidates to treat the post-acute sequelae of SARS-CoV-2 infection (PASC).

The study was well designed and performed. The manuscript is well written and provides a large amount of data supporting the deleterious impact of persistent hyperglycemia in PASC and a potential beneficial effect of vaccination during acute infection. These findings are very important and significant.

I do have few comments to further improve the quality of the manuscript, as listed below:

Main Comments:

1) SARS-CoV-2-infected AGM was previously described as a model of severe Covid. As Long-Covid occurs more often in people who had severe disease, this supports the similarities found between this NHP model and the pathology in humans. A question regarding the viral strain used to infect these AGMs: in the present study, the authors infected animals with the strain 2019-nCoV/USA-WA1/2000 that circulated in the first months of the Covid-19 pandemics. Do the authors believe that this viral strain is determinant to the establishment of Long-Covid in AGMs? Or the same outcome could be observed in animals infected with other variants that became predominant (less pathogenic/more adapted) overtime?

2) The authors showed that long-term hyperglycemia is a hallmark of SARS-CoV-2 infection in AGMs. The systemic effects of persistent dysregulated glucose metabolism associates with an inflammatory signature characterized by chemokines and some inflammatory molecules. This

proteomic profile was assessed by using the Olink proximity extension assay. The authors reported that among the 96 proteins tested, 65 were analyzed following stringent data QC. Other cellular pathways were described to play a key role in the SARS-CoV-2 pathogenesis and inflammation (interferons and other pro-inflammatory cytokines, for instance). Were these proteins included in the 96-panel and excluded following QC or they were not considered in the analyses? Please comment.

3) The authors analyzed the persistence of replicating virus by different techniques in the duodenum, liver and pancreas of infected animals at week 18. Did the authors investigate the presence of replicating virus in other tissues at later time points, such as lungs or olfactory bulbs, that could be associated with symptoms currently observed in Long-Covid?

4) In the analyses of T cell hypersensitivity to PMA/iono stimulation, the authors found correlations of naïve and memory CD4 populations with glucose or GDNF levels in plasma. In general, naïve and memory cells rely more in a OXPHOS metabolism whereas effector cells are more likely to be driven by glycolysis for their energetic needs upon activation. Here these “naïve” cells express markers of activation and cytokine production. Could the authors comment whether these results were expected in a set of polyclonal stimulation and if these cells can still be classified as naïve?

Minor Comments:

1) The rationale to investigate the impact of vaccination at day 4 post-infection on glucose metabolism rather than vaccination prior to SARS-CoV-2 infection should be better explained in the introduction section.

2) The authors referred to the blood glucose levels in non-infected male and female AGMs. Do the authors have this data following vaccination with BNT162b2 in non-infected AGMs?

3) Typos:

Lane 226 – CCI25 > L

Lane 299 – SAR-CoV-2 > S

Reviewer #3 (Remarks to the Author):

This manuscript from Palmer and colleagues describes interesting, novel studies on hyperglycemia following Wuhan Hu-1 infection in African Green Monkeys. It makes some helpful additions to the field for those trying to map the relationship between Long Covid and metabolic disease. Their central point is well and clearly made: as discussed from line 146, they find a persistent hyperglycemia maintained to 12 weeks. This indeed makes it a useful model to pursue. If I had reservations about the manuscript they related to the loose joining up of narrative and causality between the metabolic analysis and the rather dissociated contemporaneous immunological analysis, the former rather poorly illuminated by the latter. The title refers to

'immune correlates of hyperglycemia', but the term 'correlates' has to do some heavy lifting here!
I have the following specific feedback for the authors:

1. At line 26: hyperglycemia is described as a 'major manifestation' of PASC. While it has certainly been described, 'major manifestation' would be an overstatement from the current literature. To give the reader a clearer context, the authors may wish to refer to an cite:

- Xie Y, Al-Aly Z. Risks and burdens of incident diabetes in long COVID: a cohort study. *Lancet Diabetes Endocrinol.* 2022 May;10(5):311-321. doi: 10.1016/S2213-8587(22)00044-4. Epub 2022 Mar 21. PMID: 35325624; PMCID: PMC8937253.

- Altmann DM, Whettlock EM, Liu S, Arachchillage DJ, Boyton RJ. The immunology of long COVID. *Nat Rev Immunol.* 2023 Oct;23(10):618-634. doi: 10.1038/s41577-023-00904-7. Epub 2023 Jul 11. Erratum in: *Nat Rev Immunol.* 2023 Sep 18;; PMID: 37433988.

2. The authors need to find consistency on their use of PASC versus Long Covid as a term; internationally many now favour the latter term

3. At line 79 we zoom into results, but its actually hard to get any sense of what was done without constantly cross-checking the Methods since so little information is given. It would have been helpful to have mentioned numbers, ± vaccination, infecting strain and dose - otherwise the reader has to work quite hard to uncover all of this.

4. Many (immunologist) readers would find the framing of the PMA/ionomycin T cell studies a little perplexing. This is simply a basic lab protocol to polyclonally activate cells. It isn't normally regarded as a special assay for 'inflammatory responses' or 'hypersensitivity'. This is rather misleading as an interpretation. For example, the statement at line 254 was hard to decode.

5. The point (line 160) that vaccination has a protective effect for hyperglycemia is important and perhaps merits more attention.

6. The authors occasionally attempt to link specific observations to human acute infection but have not attempted to do so in the most obvious example since acute O-LINK data are available for the huge PHOSP cohort. (<https://www.medrxiv.org/content/10.1101/2023.06.07.23291077v1>)
How well do these align?

7. An aspect of Long Covid where these findings differ from human studies is in the failure to find any persistent virus in tissues. This merits further comment as to the causes of the difference. They may want to relate to human studies such as Zollner's Gastroenterology paper on the relationship between gut reservoirs and Long Covid, or Stein's Nature paper on tissue autopsy findings:

Stein SR, Ramelli SC, Grazioli A, Chung JY, Singh M, Yinda CK, Winkler CW, Sun J, Dickey JM, Ylaya K, Ko SH, Platt AP, Burbelo PD, Quezado M, Pittaluga S, Purcell M, Munster VJ, Belinky F, Ramos-Benitez MJ, Boritz EA, Lach IA, Herr DL, Rabin J, Saharia KK, Madathil RJ, Tabatabai A, Soherwardi S, McCurdy MT; NIH COVID-19 Autopsy Consortium; Peterson KE, Cohen JI, de Wit E, Vannella KM, Hewitt SM, Kleiner DE, Chertow DS. SARS-CoV-2 infection and persistence in the human body and brain at autopsy. *Nature.* 2022 Dec;612(7941):758-763. doi: 10.1038/s41586-022-05542-y.

Reviewer #1:

The manuscript by Palmer and colleagues investigates hyperglycemia in a non-human African Green Monkey (AGM) primate model of acute, short-term, and Long COVID or Post Acute Sequelae of COVID-19 (PASC). Additionally, the effect of vaccination during the acute phase of SARS-CoV-2 infection on glucose regulation long-term. The investigated infected 15 AGMs with 1×10^6 TCID₅₀ of the SARS-CoV-2 virus 2019-nCoV/USA-WA1/2020 through the intranasal and intratracheal route. Five of the animals received one dose of the original Pfizer/BioNTech BNT162b2 vaccine 4 days post infection. In life samples were collected throughout the study time course over 18 weeks. Samples were evaluated for viral load, serum cytokine responses, antibody responses, T cell activation, and insulin regulation. Viral RNA was found to be retained in the tissues of infected animals even 18-weeks post infection while viral RNA shedding in the upper respiratory tract had mainly cleared by week 5. The authors found that dysregulation of chemokines and hypersensitivity of T cell responses which correlated with elevated and persistent hyperglycemia at least 4 months post SARS-CoV-2 infection. GDNF increased in the non-vaccinated animals 103 weeks post infection compared to baseline which correlated with the chemokine CCL25. Interestingly, polyfunctional CD4 T cells numbers correlated with plasma glucose and GDNF levels. Glucose regulation was stabilized in animals that received a COVID-19 vaccine. In summary the authors show a potential non-human primate model of metabolic PASC. Currently, there is a lack of preclinical models which can be used to understand the mechanisms promoting Long COVID. Additionally, there is little understood about the long-term regulation of insulin and glucose following COVID-19 and how glucose regulation may play a role in PASC. The work put forth by the authors is of interest to the field as it both puts forth a potential preclinical model of PASC and offers novel findings implicating the dysregulation of glucose months following SARS-CoV-2 infection suggesting a role for glucose regulation and chemokines in the development and maintenance of Long COVID.

We appreciate the overall favorable comments from the reviewer on how our preclinical model of metabolic PASC can "offers novel findings implicating the dysregulation of glucose".

1. Results – Figure of study design. Figure 1a was poorly described in the results section and in the introduction. Can the authors please add a description of the study including the number of animals used, the infectious dose, route of infection, the vaccine administered, the type and number of samples collected, and specifically when the samples were collected? It was difficult to evaluate the study's results without this information. Especially missing was the description of the samples collected. It was unclear if the antibody analysis was from mucosal or blood samples.

We thank the reviewer for these thoughtful comments and suggestions which have been addressed (Pages 4, 5; Lines 95-109).

2. Study Design Rationale. It was not clear or described why the authors decided to vaccinate the animals 4 days post infection. This study question would have been important when vaccines were first rolled out as it was unknown if vaccination in a vulnerable person during the acute phase of SARS-CoV-2 infection could decrease disease severity. However, at this time, since so many people have pre-existing immunity through vaccination or infection, it is unclear why vaccination would have been performed after infection. Can the author's please clarify this?

We agree with this important reasoning by the reviewer. However, although the question of whether COVID-19 severity would be decreased if vaccine is administered during acute infection is valid, the current study focuses on the impact of longer-term infection on metabolic disease. Since tissue (e.g., metabolic tissues - liver, pancreas) inflammatory processes against viral infection may aggravate glucose homeostasis. Thus, we reasoned that mRNA vaccines which may elicit an immune response within hours may offer benefits to tissues prior to multiorgan distribution of SARS-CoV-2 produced by viral shedding from the lungs into body fluids. This is articulated on Page 4, Lines 79-85.

3. Results – Figure clarity. Many of the figures were too small to read especially in Figures 1b, 2b, 3a, 3d, 3e, 4a, 4g, 4h, 5b, and 5d. Can the authors please increase the size of the fonts? In Figure 4a, I can't read any of the text in yellow. Can this please be corrected? Additionally, there are 2 panels labeled Figure 2f. This was confusing. Can Figure 2 be better presented?

We thank the reviewer for bringing this to our attention. All figures have been updated to address these concerns.

4. Results – Figure Labeling. The graphs often do not indicate what group the samples came from – vaccinated or unvaccinated. Can the authors please add this to the graphs to make it easier to read? For example in Figure 4. We thank the reviewer for this suggestion which has been addressed.

5. Results – Supplementary Tables. The supplementary Tables have a formatting issue where there is overlapping text. Can the authors please fix this in Table S1 and Table S2? We thank the reviewer for highlighting this observation. We re-checked the formatting of the uploaded version.

6. Results – Additional Baseline Values. The authors are evaluating Glucose, Triglycerides, and cholesterol over 18 weeks, but there are only one or two baseline values. Can the authors provide a long-term measurement of these parameters as a comparison to the data from SARS-CoV-2 infected animals?

We thank the reviewer for this important suggestion. We would like to highlight that glucose readings at pre-baseline were taken when the animals were in the general colony. Animals were moved to a controlled research environment during this study (including BL1 and BL2), and so ideally it would be best to compare glucose levels post infection with those taken at baseline (BL1 = baseline 1 (6.5 weeks pre-infection); BL2 = baseline 2 (1.5 weeks pre-infection). With the reviewer’s suggestion we were still curious to determine the robustness of our results. Hence, we examined the pre-baseline glucose levels in blood collected every few months (December 2018 - November 2021) in animals housed in the colony. Please note that baseline samples were collected in January 2022 (BL1) and February 2022 (BL2). The data is presented in Fig. S3 and Page 8-9, Lines 180-188. Overall, the cumulative glucose levels post- SARS-CoV-2 infection were significantly higher than the composite pre-baseline readings in the unvaccinated group.

The biochemical analyses performed on colony animals are more limited than those conducted for this specific research. Thus, unfortunately the triglyceride or cholesterol data pre-baseline are unavailable for this project. Moreover, we focused on hyperglycemia for this manuscript since the changes in the glucose levels were more profound than those of triglycerides and cholesterol.

7. Study design. Authors should explain the sex imbalance of the animals used in the study.

There is an overabundance of females in the study based on the limited availability of animals that we can assign. Normally we would seek to have similar numbers of males and females, however, as sometimes is the case, this is not always possible with NHPs as they are a limited resource. This is especially true for AGMs, which we do not breed and have previously imported these wild-caught animals. We have noted this as a potential cofounder in the discussion (Page 21, Lines 483-485). We have also included a statement in the method’s section to explain the overabundance of females in this study (Page 23, Line 517-520).

Reviewer #2:

In this study, Palmer and colleagues characterized the pathogenesis of SARS-CoV-2-infection in African green monkeys (AGMs), mostly focusing in the systemic impact of glucose metabolism and inflammation. They reported that this non-human primate (NHP) species can be used as an animal model to assess the impact of Long-Covid and to test therapeutic candidates to treat the post-acute sequelae of SARS-CoV-2 infection (PASC). The study was well designed and performed. The manuscript is well written and provides a large amount of data supporting the deleterious impact of persistent hyperglycemia in PASC and a potential beneficial effect of vaccination during acute infection. These findings are very important and significant. I do have few comments to further improve the quality of the manuscript, as listed below:

We appreciate the favorable appraisal of our study, and that the reviewer found the manuscript well-written and important.

Main Comments:

1) SARS-CoV-2-infected AGM was previously described as a model of severe Covid. As Long-Covid occurs more often in people who had severe disease, this supports the similarities found between this NHP model and the pathology in humans. A question regarding the viral strain used to infect these AGMs: in the present study,

the authors infected animals with the strain 2019-nCoV/USA-WA1/2000 that circulated in the first months of the Covid-19 pandemics. Do the authors believe that this viral strain is determinant to the establishment of Long-Covid in AGMs? Or the same outcome could be observed in animals infected with other variants that became predominant (less pathogenic/more adapted) overtime?

We thank the reviewer for this important question. Since other SARS-CoV-2 strains also perpetuate inflammation, we would expect similar inflammatory-mediated glycometabolic disturbances. In fact, studies in humans have shown that beyond the wild-type strain, Alpha, Delta, and Omicron, can induce long COVID-19 symptoms (Du et al., 2022; PMID: 36498103), with no significant differences in the prevalence of the most common symptoms (Aloe et al., 2023, PMID: 37511933). In addition, it has been shown that gamma, delta, and omicron variants, can stimulate hepatic glucose production and disposal through increased activity of the rate limiting gluconeogenic enzyme phosphoenolpyruvate carboxykinase (Barreto et al., 2023; PMID: 37186819). Please see this articulated in discussion (Page 17, Lines 392-395).

2) The authors showed that long-term hyperglycemia is a hallmark of SARS-CoV-2 infection in AGMs. The systemic effects of persistent dysregulated glucose metabolism associates with an inflammatory signature characterized by chemokines and some inflammatory molecules. This proteomic profile was assessed by using the Olink proximity extension assay. The authors reported that among the 96 proteins tested, 65 were analyzed following stringent data QC. Other cellular pathways were described to play a key role in the SARS-CoV-2 pathogenesis and inflammation (interferons and other pro-inflammatory cytokines, for instance). Were these proteins included in the 96-panel and excluded following QC or they were not considered in the analyses? Please comment.

We appreciate this very thoughtful and significant comment. The Olink 96-inflammation panel which includes 92 target proteins, and 4 controls (internal/external) is shown below, with analytes failing QC highlighted in yellow. We agree, interferons and related proteins were underrepresented in this panel in general. We conducted STRING analysis on the proteins that failed and found them to be significantly related (FDR <0.05) to interleukin signaling (Wiki Pathways, Local Network Cluster -STRING, Reactone pathway, Kegg pathway). We have addressed this as a limitation (Page 21, Lines 485-491).

Assay	Uniprot ID	QC
IL8	P10145	Passed
VEGFA	P15692	Passed
CD8A	P01732	Failed
MCP-3	P80098	Failed
GDNF	P39905	Passed
CDCP1	Q9H5V8	Passed
CD244	Q9BZW8	Failed
IL7	P13232	Failed
OPG	O00300	Passed
LAP TGF-beta-1	P01137	Passed
uPA	P00749	Passed
IL6	P05231	Passed
IL-17C	Q9P0M4	Failed
MCP-1	P13500	Passed
IL-17A	Q16552	Failed
CXCL11	O14625	Passed
AXIN1	O15169	Passed
TRAIL	P50591	Passed
IL-20RA	Q9UHF4	Passed
CXCL9	Q07325	Passed

CST5	P28325	Passed
IL-2RB	P14784	Failed
IL-1 alpha	P01583	Failed
OSM	P13725	Passed
IL2	P60568	Failed
CXCL1	P09341	Passed
TSLP	Q969D9	Failed
CCL4	P13236	Passed
CD6	P30203	Passed
SCF	P21583	Passed
IL18	Q14116	Passed
SLAMF1	Q13291	Passed
TGF-alpha	P01135	Passed
MCP-4	Q99616	Passed
CCL11	P51671	Passed
TNFSF14	O43557	Passed
FGF-23	Q9GZV9	Passed
IL-10RA	Q13651	Failed
FGF-5	P12034	Passed
MMP-1	P03956	Passed
LIF-R	P42702	Passed
FGF-21	Q9NSA1	Passed
CCL19	Q99731	Passed
IL-15RA	Q13261	Failed
IL-10RB	Q08334	Passed
IL-22 RA1	Q8N6P7	Failed
IL-18R1	Q13478	Passed
PD-L1	Q9NZQ7	Passed
Beta-NGF	P01138	Failed
CXCL5	P42830	Failed
TRANCE	O14788	Passed
HGF	P14210	Passed
IL-12B	P29460	Passed
IL-24	Q13007	Failed
IL13	P35225	Failed
ARTN	Q5T4W7	Failed
MMP-10	P09238	Passed
IL10	P22301	Failed
TNF	P01375	Passed
CCL23	P55773	Passed
CD5	P06127	Passed
CCL3	P10147	Passed
Flt3L	P49771	Passed
CXCL6	P80162	Passed
CXCL10	P02778	Passed
4E-BP1	Q13541	Passed

IL-20	Q9NYY1	Failed
SIRT2	Q8IXJ6	Passed
CCL28	Q9NRJ3	Failed
DNER	Q8NFT8	Passed
EN-RAGE	P80511	Failed
CD40	P25942	Passed
IL33	O95760	Failed
IFN-gamma	P01579	Failed
FGF-19	O95750	Passed
IL4	P05112	Failed
LIF	P15018	Failed
NRTN	Q99748	Failed
MCP-2	P80075	Passed
CASP-8	Q14790	Passed
CCL25	O15444	Passed
CX3CL1	P78423	Passed
TNFRSF9	Q07011	Passed
NT-3	P20783	Passed
TWEAK	O43508	Passed
CCL20	P78556	Passed
ST1A1	P50225	Passed
STAMBP	O95630	Passed
IL5	P05113	Failed
ADA	P00813	Passed
TNFB	P01374	Passed
CSF-1	P09603	Passed

3) The authors analyzed the persistence of replicating virus by different techniques in the duodenum, liver and pancreas of infected animals at week 18. Did the authors investigate the presence of replicating virus in other tissues at later time points, such as lungs or olfactory bulbs, that could be associated with symptoms currently observed in Long-Covid?

Yes, indeed we conducted qPCR analysis on lungs samples at necropsy and found no signals. We added this important information (Page 13, Line 305).

4) In the analyses of T cell hypersensitivity to PMA/iono stimulation, the authors found correlations of naïve and memory CD4 populations with glucose or GDNF levels in plasma. In general, naïve and memory cells rely more in a OXPHOS metabolism whereas effector cells are more likely to be driven by glycolysis for their energetic needs upon activation. Here these “naïve” cells express markers of activation and cytokine production. Could the authors comment whether these results were expected in a set of polyclonal stimulation and if these cells can still be classified as naïve?

We thank the reviewer for this important comment. We have missed important cell populations when we originally set the threshold population number to 15 (condensing to 9) during the Spectre analysis. We revised this setting the threshold number to 30 (condensing to 18). We found activated CD8+ T cells percentages and numbers to be significantly associated with glucose and GDNF levels (updated results on page 12, Lines 272-293 and Fig 5, and FigS5). Some activated memory CD4 T cell populations were associated with glucose levels but did not reach the level of significance as that of activated CD8 T cells.

Conventionally, naive and memory CD4 T cells have been considered non-activated and “resting”. However recent advances in immunometabolism have now shown that in the context of infection/activation, some naive and memory CD4 T cells may be metabolically active and functional. For example, HIV susceptibility between

naive and more differentiated subsets are linked to their metabolic activity (oxidative phosphorylation and glycolysis), independent of canonical markers of differentiation and activation phenotype (PMID: 30581119). Thus, during PMA/I stimulation, some cells expressing markers of naive and memory T cells is likely to express activation markers and produce cytokines.

Minor Comments:

1) The rationale to investigate the impact of vaccination at day 4 post-infection on glucose metabolism rather than vaccination prior to SARS-CoV-2 infection should be better explained in the introduction section.

A rationale for vaccination post infection is given on Page 4, Line 79-85.

2) The authors referred to the blood glucose levels in non-infected male and female AGMs. Do the authors have this data following vaccination with BNT162b2 in non-infected AGMs?

We do not have this data. The key purpose of the vaccination arm of the study was to evaluate the potential impact of the vaccine on immunological parameters and long-term COVID-19 manifestations. It is indeed unclear whether or how vaccine-induced responses could impact long-term disease pathology. A small prospective study aimed to shed light on this found vaccination generally resulted in more favorable health outcomes.

3) Typos:

Lane 226 – CCI25 > L; Lane 299 – SAR-CoV-2 > S

We regret the oversight. These typos have been corrected.

Reviewer #3

This manuscript from Palmer and colleagues describes interesting, novel studies on hyperglycemia following Wuhan Hu-1 infection in African Green Monkeys. It makes some helpful additions to the field for those trying to map the relationship between Long Covid and metabolic disease. Their central point is well and clearly made: as discussed from line 146, they find a persistent hyperglycemia maintained to 12 weeks. This indeed makes it a useful model to pursue.

We appreciate the reviewer favorable appraisal of our work with respect to the validity of pursuing SARS-CoV-2 infected AGMs as a model for studying COVID-19 related long-term metabolic diseases.

If I had reservations about the manuscript they related to the loose joining up of narrative and causality between the metabolic analysis and the rather dissociated contemporaneous immunological analysis, the former rather poorly illuminated by the latter. The title refers to 'immune correlates of hyperglycemia', but the term 'correlates' has to do some heavy lifting here!

We have changed 'correlates' to 'associates' in the title.

I have the following specific feedback for the authors:

1) 'major manifestation' would be an overstatement from the current literature. To give the reader a clearer context, the authors may wish to refer to an cite:

- Xie Y, Al-Aly Z. Risks and burdens of incident diabetes in long COVID: a cohort study. *Lancet Diabetes Endocrinol.* 2022 May;10(5):311-321. doi: 10.1016/S2213-8587(22)00044-4. Epub 2022 Mar 21. PMID: 35325624; PMCID: PMC8937253.
- Altmann DM, Whettlock EM, Liu S, Arachchillage DJ, Boyton RJ. The immunology of long COVID. *Nat Rev Immunol.* 2023 Oct;23(10):618-634. doi: 10.1038/s41577-023-00904-7. Epub 2023 Jul 11. Erratum in: *Nat Rev Immunol.* 2023 Sep 18; PMID: 37433988.

We thank the reviewers for these suggestions. We have toned back “major manifestation” to “a metabolic manifestation”. We have included the suggested references (Page 2, Lines 27-28)

2) The authors need to find consistency on their use of PASC versus Long Covid as a term; internationally many now favour the latter term.

Because we would like to highlight the descriptive (metabolic) syndrome of COVID (metabolic syndrome) we feel that in this specific context “metabolic PASC” would convey the intended message more appropriately. We have made this consistent throughout the manuscript.

3) At line 79 we zoom into results, but its actually hard to get any sense of what was done without constantly cross-checking the Methods since so little information is given. It would have been helpful to have mentioned numbers, \pm vaccination, infecting strain and dose - otherwise the reader has to work quite hard to uncover all of this.

We have added this information in the result section (Section: Study groups) (Page 4, Lines 95-109).

4) Many (immunologist) readers would find the framing of the PMA/ionomycin T cell studies a little perplexing. This is simply a basic lab protocol to polyclonally activate cells. It isn't normally regarded as a special assay for 'inflammatory responses' or 'hypersensitivity.' This is rather misleading as an interpretation. For example, the statement at line 254 was hard to decode.

We appreciate the reviewer's comment. We have changed “hypersensitivity” throughout the manuscript to “ex-vivo polyclonal activation profile”. We have also modified the test to address this concern (Page 7, Lines 145 - 158 and Page 12-13, Lines 272-293).

5) The point (line 160) that vaccination has a protective effect for hyperglycemia is important and perhaps merits more attention.

We have expanded on this statement with a follow up statement “and may be explored to reduce the long-term burden of new-onset COVID-19 related diabetes” (Page 9, Lines 196-197). As well, we have contextualized the reasoning behind administering the vaccine during the acute infection (Page 4, Line 79-85).

6) The authors occasionally attempt to link specific observations to human acute infection but have not attempted to do so in the most obvious example since acute O-LINK data are available for the huge PHOSP cohort. (<https://www.medrxiv.org/content/10.1101/2023.06.07.23291077v1>)

How well do these align?

This study refers to a large scale PASC study in humans utilizing the Olink Explore 384 Inflammation panel. The pattern of regulated plasma proteins did not overlap with our results. The PASC evaluated were related to fatigue, anxiety/depression, gastrointestinal and cardiorespiratory symptoms, and not of classic metabolic diseases (e.g. hyperglycemia/diabetes). Indeed, the authors found that although persistent inflammation defined PASC, distinct profiles were associated with specific disease manifestations. We have included this in our discussion (Page 21, Lines 478-482).

7) An aspect of Long Covid where these findings differ from human studies is in the failure to find any persistent virus in tissues. This merits further comment as to the causes of the difference. They may want to relate to human studies such as Zollner's Gastroenterology paper on the relationship between gut reservoirs and Long Covid, or Stein's Nature paper on tissue autopsy findings:

- Stein SR, Ramelli SC, Grazioli A, Chung JY, Singh M, Yinda CK, Winkler CW, Sun J, Dickey JM, Ylaya K, Ko SH, Platt AP, Burbelo PD, Quezado M, Pittaluga S, Purcell M, Munster VJ, Belinky F, Ramos-Benitez MJ, Boritz EA, Lach IA, Herr DL, Rabin J, Saharia KK, Madathil RJ, Tabatabai A, Soherwardi S, McCurdy MT; NIH COVID-19 Autopsy Consortium; Peterson KE, Cohen JI, de Wit E, Vannella KM, Hewitt SM, Kleiner DE, Chertow DS. SARS-CoV-2 infection and persistence in the human body and brain at autopsy. *Nature*. 2022 Dec;612(7941):758-763. Doi: 10.1038/s41586-022-05542-y.

We thank the reviewer for this important point. We have added the following to the discussion with the recommended references (Page 18, Line 403-412).

“While the persistence of viral nucleic acid and antigens in some extra pulmonary tissues have been linked to PASC in patients with underlying illnesses, there is debates on how clinically relevant these “viral ghosts” are, and the mechanisms by which they contribute to PASC. In fact, quantification of SARS-CoV-2 RNA (N) using the highly sensitive through ddPCR in autopsy liver and pancreas from 11 late case patients who died with COVID-19, yielded negative in most of the deceased samples. In the same study a significant proportion had low signals RNA (<0.1 N gene copies per nanogram input RNA) or negative in the intestine). We detected no significant levels of viral RNA in lungs, but none of our animals died of COVID-19 complications. Hence the higher viral burden detected in respiratory sites in some deceased patients may reflect a link between viral load and mortality.”

REVIEWERS' COMMENTS

Reviewer #1 (Remarks to the Author):

The authors have addressed my concerns.

Reviewer #2 (Remarks to the Author):

The authors have considered comments of the reviewers and provided very reasonable responses and corresponding modifications to the manuscript.

Reviewer #3 (Remarks to the Author):

At a time when all regulators note the dearth of any pertinent animal models for Long Covid evaluations, the work-up of models such as this assumes importance.

The authors have had a lot of feedback to the initial submission which had a fair number of weaknesses. They have amended appropriately and now have quite a useful manuscript for the field.